# Cation-induced changes in the inner- and outer-sphere mechanisms of electrocatalytic $CO_2$ reduction

Xueping Qin [1] ✉, Heine A. Hansen [1], Karoliina Honkala [2] & Marko M. Melander [2] ✉

The underlying mechanism of cation effects on $CO_2RR$ remains debated. Herein, we study cation effects by simulating both outer-sphere electron transfer (OS-ET) and inner-sphere electron transfer (IS-ET) pathways during $CO_2RR$ via constrained density functional theory molecular dynamics (cDFT-MD) and slow-growth DFT-MD (SG-DFT-MD), respectively. Our results show without any cations, only OS-ET is feasible with a barrier of 1.21 eV. In the presence of $K^+$ ($Li^+$), OS-ET shows a very high barrier of 2.93 eV (4.15 eV) thus being prohibited. However, cations promote $CO_2$ activation through IS-ET with the barrier of only 0.61 eV ($K^+$) and 0.91 eV ($Li^+$), generating the key intermediate (adsorbed $CO_2^{\delta-}$). Without cations, $CO_2$-to-$CO_2^{\delta-}$(ads) conversion cannot proceed. Our findings reveal cation effects arise from short-range Coulomb interactions with reaction intermediates. These results disclose that cations modulate the inner- and outer-sphere pathways of $CO_2RR$, offering substantial insights on the cation specificity in the initial $CO_2RR$ steps.

The electrocatalytic $CO_2$ reduction reaction ($CO_2RR$) is a promising technology to impact carbon emissions and store excess renewable energy in chemicals and fuels[1,2]. Developing electrocatalysts achieving high $CO_2RR$ activity and selectivity is of paramount importance to move from laboratory-level demonstrations to industrial, large-scale $CO_2RR$ implementations. Among various metal electrocatalysts working in aqueous media, gold and silver exclusively catalyze $CO_2$ producing carbon monoxide[3], while on copper various products can be obtained including methane, ethylene, and different alcohols[4]. Substantial efforts have been devoted to steering the activity and selectivity by modifying catalyst surfaces or tailoring electrode configurations[5]. However, it is not enough to understand the electrode material alone as it has been well-established that both electrolyte composition and interfacial microenvironment strongly impact $CO_2RR$ performance[6–8]. In particular, cation effects have recently gained significant attention and have been shown to substantially influence $CO_2RR$ reaction rates and selectivity[9–11] – it has even been demonstrated the $CO_2RR$ cannot proceed without cations which warrants detailed understanding of cation effects[12].

The underlying interactions or mechanisms of specific cation effects on $CO_2RR$, however, remain elusive and controversial[9,10,12–18]. As a result, several diverging theories have been proposed: (1) local electrostatics where cations modify the local electric field and change the electrostatic potential profile by specific adsorption in the electrical double layer (EDL)[19–21]; (2) cations act as interfacial pH buffers with larger cations being better buffers thus decreasing the local pH following the trend: $Li^+$>$Na^+$>$K^+$>$Cs^+$[17,18]; (3) cations stabilize key reaction intermediates, such as adsorbed $CO_2$, and thereby enhance binding thermodynamics and reaction kinetics[12,14–16,22]. The possibility of specific adsorption and partial desolvation of cations concerning $CO_2RR$ was largely ruled out[15,16,23], and it was instead suggested that cations accumulate at the outer Helmholtz plane (OHP) and stabilize reaction intermediates via local electrostatic interactions within the EDL. However, the recent work by Waegele et al. provided evidence that specific cation adsorption greatly facilitates $CO_2RR$ through

[1]Department of Energy Conversion and Storage, Technical University of Denmark, Anker Engelunds Vej Building 301, Kgs. Lyngby 2800, Denmark. [2]Department of Chemistry, Nanoscience Center, University of Jyväskylä, P.O. Box 35, FI-40014 Jyväskylä, Finland. ✉e-mail: xueqi@dtu.dk; marko.m.melander@jyu.fi

short-range interactions[24]. Furthermore, it remains unclear whether the short-range chemical or electrostatic interaction between $CO_2RR$ intermediates and (quasispecifically) adsorbed cations, or the long-range electrostatic interaction between electric field and dipole moments of adsorbed intermediates (electrostatic adsorption of cations) is the main source of cation specificity.

The detailed operational mechanism of cation effects remains unknown even for forming $CO_2^{\delta-}$ through electron transfer (ET) which is the first $CO_2RR$ step and generally accepted to be the rate-determining step[25–27]. Two very recent experimental studies attributed the cation effect in $CO_2^{\delta-}$ formation to either quasichemical interactions between $CO_2^{\delta-}$ and cations or cation-modified dipole-field interactions[12,14]. Specifically, Monteiro et al.[12] observed CO production on gold electrodes only in metal cation-containing electrolytes using cyclic voltammetry and scanning electrochemical microscopy measurements. Complementary density functional theory (DFT) simulations[12] explained this through the stabilization of an adsorbed $CO_2^-$ (ads) intermediate by partially desolvated or specifically adsorbed[24] cations with the short-range interaction, in agreement with the cation-coupled electron transfer picture[22]. Although the calculations in Ref. 12 were questioned by Le and Rahman[28], the main mechanistic conclusions still hold even if the artificial constraint of fixing the distance of the C ($CO_2$) from the electrode is removed[29]. In contrast, Gu et al.[14] ascribed the high Faradaic efficiency of CO production ( ~ 90%) on gold to electric field modulation induced by hydrated alkali cations. This was interpreted to arise from long-range dipole-field interactions stabilizing the $CO_2$(ads) species. They also proposed that such a shielded electric field prohibits the diffusion of protons thus suppressing the competitive hydrogen evolution reaction (HER)[14]. Currently, the experimental studies have not reached a consensus on the origin of cation promotion effects in $CO_2RR$.

Current theoretical/computational understanding of cation and electrolyte effects in electrocatalysis is limited to reaction thermodynamics as kinetic studies have remained scarce[16,30]. While most experimental and computational studies in particular have considered $CO_2RR$ steps *on* the electrode surface, the initial ET step is expected to take place either *during* $CO_2$ adsorption from solution forming a partially adsorbed $CO_2^{\delta-}$(ads) or *before* adsorption producing a $CO_2^-$ (sol) anion in solution. Formation of $CO_2^{\delta-}$(ads) can be seen as an inner-sphere ET (IS-ET) process, and recent computational studies showed that cations have a central role in facilitating the reaction kinetics of this step as well as the subsequent inner-sphere reduction process[31,32]. Generation of $CO_2^-$(sol) on the other hand proceeds via an outer-sphere ET (OS-ET) pathway where the long-range ET initiates $CO_2RR$ without adsorption. The OS-ET mechanism should be relevant especially on weakly binding electrodes such as gold and under high current densities where $CO_2$ transport to an electrode surface is blocked[33,34]. Indeed, recent calculations have demonstrated that alkali metal cations can induce formate production through an OS proton-coupled electron transfer (PCET) mechanism without $CO_2$ adsorption[35]. Another previous computational study has shown that $CO_2$ reduction can proceed through an electronically adiabatic OS-ET step on a gold surface in the absence of solvation[36]. This study, however, provides only part of the story since it just considered the so-called inner-sphere reorganization in the Marcus theory of electron transfer[37], leaving the role of cations and solvents for the OS-ET pathway unresolved.

More complementary theoretical simulations are urgently needed to provide the atomic-scale understanding of how and why cations impact the thermodynamics and kinetics of $CO_2$ activation during the first ET step. Herein, we provide such detailed insights by studying the electrocatalytic $CO_2$ reduction kinetics on a Au(110) electrode in the presence of water and cations. We have used advanced DFT-based molecular dynamics simulations to address cation effects on OS-ET and IS-ET thermodynamics and kinetics. By studying $CO_2$-to-$CO_2^-$ conversions over Au in pure water and in the presence of alkali metal cations (AM+, including K+ and Li+), we show that cations efficiently modulate the IS- and OS-ET mechanisms, thermodynamics, and kinetics of initial $CO_2RR$ at electrode-electrolyte interfaces. In the absence of cations, only the OS pathway producing $CO_2^-$(sol) is possible but with a rather high barrier (1.21 eV) whereas the IS pathway to $CO_2^{\delta-}$(ads) is highly promoted by K cations (barrier = 0.61 eV). We also demonstrate that the positive cation effect on IS-ET kinetics arises from an ionic coordination bond between $CO_2^{\delta-}$ and partially desolvated K+. As the surface charge and the reaction environment are similar for both IS- and OS-ET, the strong positive cation effect of K+ results from explicit $CO_2^{\delta-}$−K+ coordination rather than the long-range electrostatic interactions. Additionally, Li+ shows a similar cation modulation effect on $CO_2$ activation, enhancing the IS-ET with a much smaller kinetic barrier of 0.91 eV compared to the OS-ET pathway (4.15 eV). Overall, our results provide crucial atomic-scale insights into the cation effects in elementary $CO_2RR$ kinetics and thermodynamics, and show that the electrolyte can control ($CO_2RR$) electrocatalysis through quasispecific chemical interactions.

## Results

We focus on the first $CO_2$ activation step initiating the overall electrocatalytic reaction[12–14] either through IS-ET to form $CO_2^{\delta-}$(ads) on the surface or OS-ET in the solution leading to $CO_2^-$(sol) which may subsequently diffuse to the surface as shown schematically in Fig. 1. We

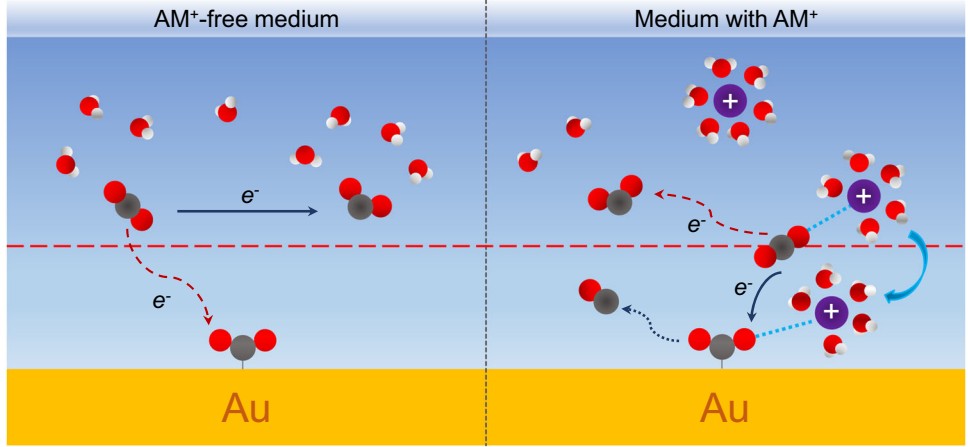

**Fig. 1 | Schematic illustration of $CO_2RR$ at Au-water interfaces.** $CO_2RR$ in AM+-free medium and medium with AM+ are shown on the left and right, respectively. The parallel red-dashed line represents the boundary of OS-ET and IS-ET.

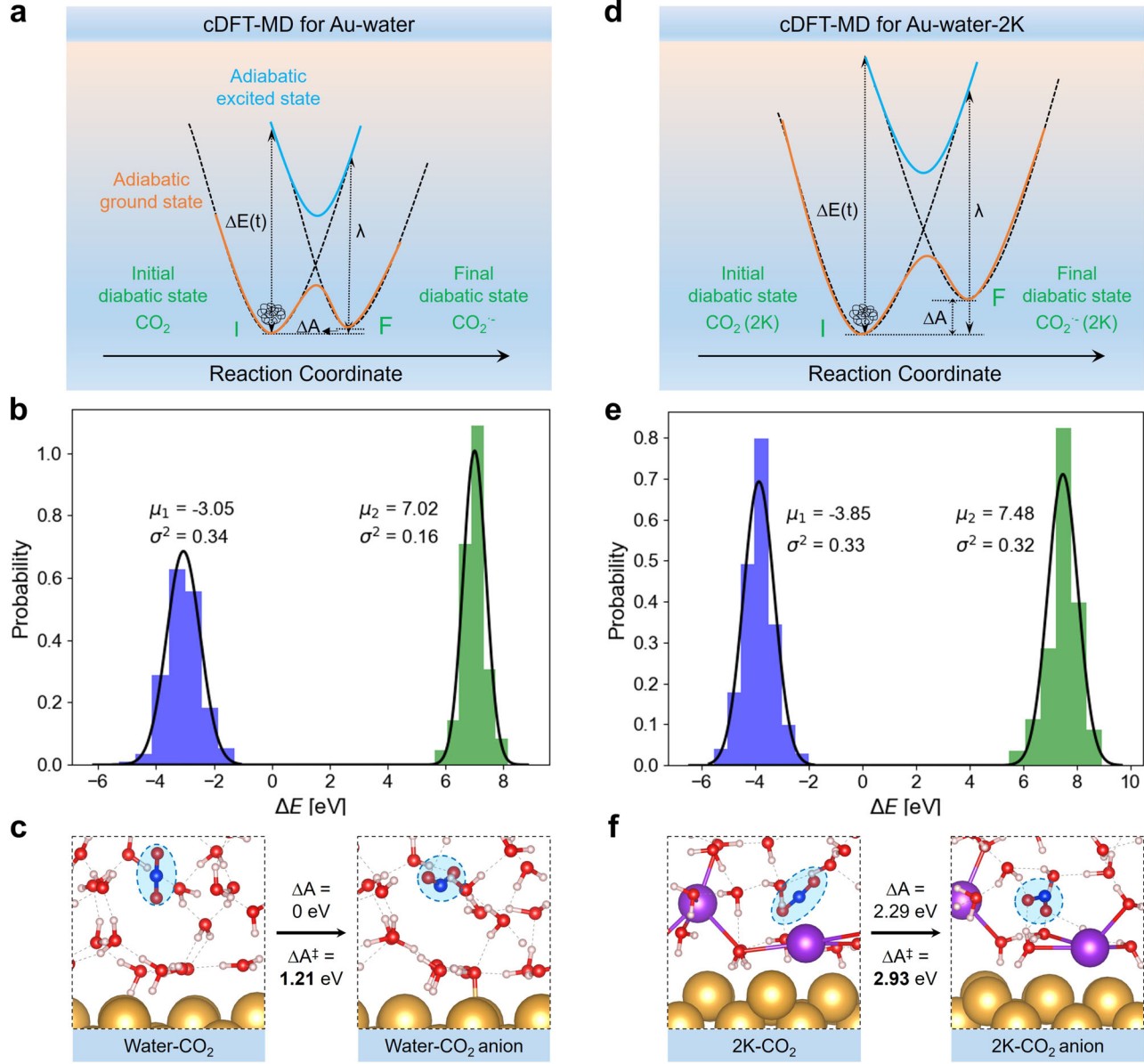

**Fig. 2 | OS-ET reactions via cDFT-MD simulations.** The schematic view of the Marcus theory, energy gap distributions, and key structures of diabatic states of OS-ET are shown at Au-water (**a**–**c**) and Au-water-2K (**d**–**f**) interfaces. In (**a**) and (**d**), the orange (blue) solid line is the adiabatic ground (excited) state while the black dashed lines are the two diabatic states (I and F) describing the initial and final states of $CO_2$ and $CO_2^-$, respectively.

compute and compare the reaction thermodynamics and kinetics of these two pathways in $AM^+$-free and $AM^+$-containing aqueous solutions. The interfacial cation concentration used is approximately 2.3M, which would correspond to a bulk concentration of roughly 0.1M–0.5M, as predicted by simulations of cation accumulation at the electrode interface under $CO_2$RR-relevant conditions[38]. While the cation concentration of 0.1M–0.5M is typical for $CO_2$RR, it has been shown that even cation concentrations as low as 0.001M can impact $CO_2$RR activity[12]. However, current DFT simulations are unable to model such low concentrations. It should be mentioned that cations and $CO_2/CO_2^-$ do not retain their full bulk solvation shell due to the high surface cation concentration.

The OS-ET was simulated using constrained density functional theory molecular dynamics (cDFT-MD) to parameterize the Marcus theory of electron transfer kinetics. As shown in Fig. 2 and discussed in the Methods, Marcus theory describes the reaction kinetics along a reorganization coordinate using diabatic, charge-localized states. Neither solvent/medium reorganization reaction coordinate nor the

required diabatic states can be captured by conventional geometric reaction coordinates or DFT methods, respectively[39]. Instead, the used cDFT method can both build diabatic states and accurately parametrize Marcus theory[40]. It is important to note that there is no universally accepted method for defining a diabatic state. The cDFT method falls within a category of empirical valence bond methods, where the definition of diabatic states relies on chemical intuition. Here we consider the cDFT diabatic description of $CO_2^-$ to be reasonably accurate, as evidenced by the computed vibrational frequencies of $CO_2^-$ closely matching experimental values (Table 1). The IS-ET was explored with more traditional geometry-based enhanced sampling methods. Specifically, we used slow-growth DFT-MD (SG-DFT-MD) to compute reaction rates within the adiabatic transition state theory.

## OS-ET pathway

The outer-sphere ET mechanism is studied by constructing two diabatic electronic states for $CO_2$ and $CO_2^-$ using cDFT as depicted in Fig. 2a and d. The OS-ET kinetics and thermodynamics are then

**Table 1 | Summary of cDFT-MD and DFT-MD simulation analysis, including C-O bond length, O-C-O angle, and simulated vibration wavenumbers with bending, symmetric stretch (denoted as symm. stret), and asymmetric stretch (denoted as asymm. stret)**

| Systems | CO$_2$ geometry | | | Vibration wavenumber (cm$^{-1}$) | | |
|---|---|---|---|---|---|---|
| | C-O1 (Å) | C-O2 (Å) | Angle (°) | Bending | Symm. stret | Asymm. stret |
| Water-CO$_2$ | 1.19 (± 0.02) | 1.19 (± 0.02) | 174 (± 3) | 615 | / | 2242/2347 |
| Water-CO$_2^-$ | 1.26 (± 0.02) | 1.25 (± 0.02) | 136 (± 4) | 673 | 1222 | 1709 |
| 2K-CO$_2$ | 1.19 (± 0.02) | 1.19 (± 0.02) | 174 (± 3) | 575 | / | 2333 |
| 2K-CO$_2^-$ | 1.28 (± 0.03) | 1.29 (± 0.03) | 125 (± 4) | 759 | 1177 | 1479 |
| 2Li-CO$_2$ | 1.19 (± 0.02) | 1.19 (± 0.02) | 174 (± 3) | 557 | / | 2378 |
| 2Li-CO$_2^-$ | 1.29 (± 0.03) | 1.28 (± 0.03) | 126 (± 5) | 712 | 1175 | 1510 |
| 2K-CO$_2^-$ (ads) | 1.26 (± 0.02) | 1.27 (± 0.02) | 126 (± 4) | 535 | 1250 | 1680 |
| 2Li-CO$_2^-$ (ads) | 1.27 (± 0.02) | 1.28 (± 0.03) | 124 (± 3) | 652 | 1032/1437 | 1765 |
| NIST-CO$_2$[a] | 1.16 | 1.16 | 180 | 667 | 1333 | 2349 |
| NIST-CO$_2^-$ | 1.25 | 1.25 | / | 714 | 1254 | 1658 |
| SERS − Ag − CO$_2^-$ [b] | / | / | / | 718 | 1130 | 1540 |

NIST and experimental data are from the references shown at bottom of the table.
[a]NIST data can be used as reference[77].
[b]In situ surface-enhanced Raman spectroscopic (SERS) data are collected on Ag electrodes[78].

computed using linear response Marcus theory (see Methods section). The needed reorganization ($\lambda$) and reaction energies ($\Delta A$) are computed from the energy gap ($\Delta E$) distributions. The energy gap is determined by calculating the instantaneous energy difference between the CO$_2$ and CO$_2^-$ states across different geometries and solvent environments with calculations performed every 50 fs. The $\Delta E$ distributions are obtained from 10 ps cDFT-MD simulations on CO$_2$ and CO$_2^-$ states in both Au-water and Au-water-2K systems as shown in Fig. 2b and e. Because CO$_2$ and CO$_2^-$ states are asymmetric (i.e., the Marcus parabolae have different curvatures) due to different solvation energies, both states are sampled and the energy gap distributions are extracted from 200 structures along the cDFT-MD trajectories (Fig. 2b, c). It should be mentioned that longer trajectories and sampling more structures could improve the statistics but already the obtained Gaussian distribution of the energy gap (as required by the microscopic Marcus theory[41]) tells that the sampling is robust (Fig. 2b, e). Furthermore, the calculations assume a local equilibrium[42] for the cation and CO$_2$ concentration profiles.

In the Au-water system, the reorganization energy for the CO$_2 \rightleftarrows$ CO$_2^-$ reaction is 4.82 eV and the reaction energy is 0.00 eV. The key structures for this step are shown in Fig. 2c including the initial and final diabatic states (water-CO$_2$ and water-CO$_2^-$). The computed reorganization energy is of similar magnitude as obtained in previous simulations for the O$_2$-to-O$_2^-$ conversion[43,44]. The reaction energy shows this reaction step is thermodynamically neutral. It should be noted that the reorganization energy measures the non-equilibrium solvent state, while the reaction energy evaluates the equilibrium solvent state. From the reorganization and reaction free energies, the OS-ET kinetics is estimated by computing the reaction barrier through Marcus theory; this gives a barrier of 1.21 eV for the CO$_2$-to-CO$_2^-$ conversion in pure water. The Marcus energy barrier of 1.21 eV indicates that OS-ET producing CO$_2$ anion in water could occur but with a slow reaction rate (normally the barrier of 0.75 eV is considered to be a threshold for facile kinetics corresponding to 1 TOF/s)[45].

To explore the cation effect, cDFT-MD simulations are repeated in the presence of K$^+$ (Au-water-2K) and Li$^+$ (Au-water-2Li). For Au-water-2K, the corresponding structure sampling and energy gap calculations are illustrated in Fig. 2d and e, and the key diabatic structures are shown in Fig. 2f. With K cations, the reorganization energy is much higher (6.30 eV) than that in the Au-water system (4.82 eV). Additionally, the reaction energy is highly endothermic being 2.29 eV, which makes the reaction very unfeasible. These lead to a very high OS-

ET barrier, 2.93 eV, making CO$_2$-to-CO$_2^-$ conversion extremely slow and improbable at Au-water-2K interfaces. In the case of Li$^+$, the prohibition of OS-ET is more pronounced than K$^+$ with even higher reorganization energy (7.60 eV) and reaction energy (3.63 eV), and thus a higher Marcus barrier (4.15 eV) as illustrated in Supplementary Fig. S1. Such a difference in OS-ET kinetics between Li$^+$ and K$^+$ is ascribed to their different hydration properties[46], where Li$^+$ is strongly hydrated compared to K$^+$ thus leading to higher water reorganization energy and resultant Marcus barrier. It should be mentioned that the OS-ET to form the CO$_2^-$ leads to less than 0.3 eV changes in the interface work function, as shown in Supplementary Table S1, which suggests the OS-ET is modeled at a nearly constant potential.

These results highlight that alkali metal cations have a very large impact on the initial OS-ET process in CO$_2$RR. When comparing water, K, and Li case, we observe that the reaction energy of CO$_2$-to-CO$_2^-$ conversion is 0.00, 2.29, and 3.63 eV, respectively, which is affected by cations and surrounding solvation structures. Without cations in the Au-water system, the reaction energy of 0.00 eV makes sense since the CO$_2$ and CO$_2^-$ configurations experience relatively fewer constraints or restrictions compared to the cases involving K and Li cations. The comparison of reaction barriers with (2.93 and 4.15 eV for K$^+$ and Li$^+$, respectively) and without (1.21 eV) cations shows that OS-ET reaction is highly prohibited by cations due to the high reorganization energy; this indicates that cations cannot activate CO$_2$ in the outer-sphere CO$_2$RR pathway. This also demonstrates that OS-ET is a plausible pathway in low-concentration electrolytes or if CO$_2$ transport is prohibited by a very dense electrolyte layer near the electrode, but the rate is expected to be low.

## IS-ET pathway

The simulated initial CO$_2$RR elementary step corresponds to a concerted, adiabatic IS-ET process, where ET and adsorption take place simultaneously leading to an activated CO$_2^{\delta-}$(ads) species on the Au surface. The cation effects are studied by simulating the CO$_2$-to-CO$_2^{\delta-}$(ads) conversion at Au-water, Au-water-2K, and Au-water-2Li interfaces by integrating the free energy profile using SG-DFT-MD.

Fig. 3a shows that at the Au-water interface the free energy keeps increasing as CO$_2$ approaches the surface, and no transition state (TS) or a thermodynamically stable final state could be identified; the CO$_2$ does not adsorb on Au(110) without cations. Two processes for the CO$_2$ approaching can nevertheless be identified: diffusion (4.25–7.35 Å) and the nearby activation (2.00–4.25 Å) as shown in Fig. 3a. The Bader

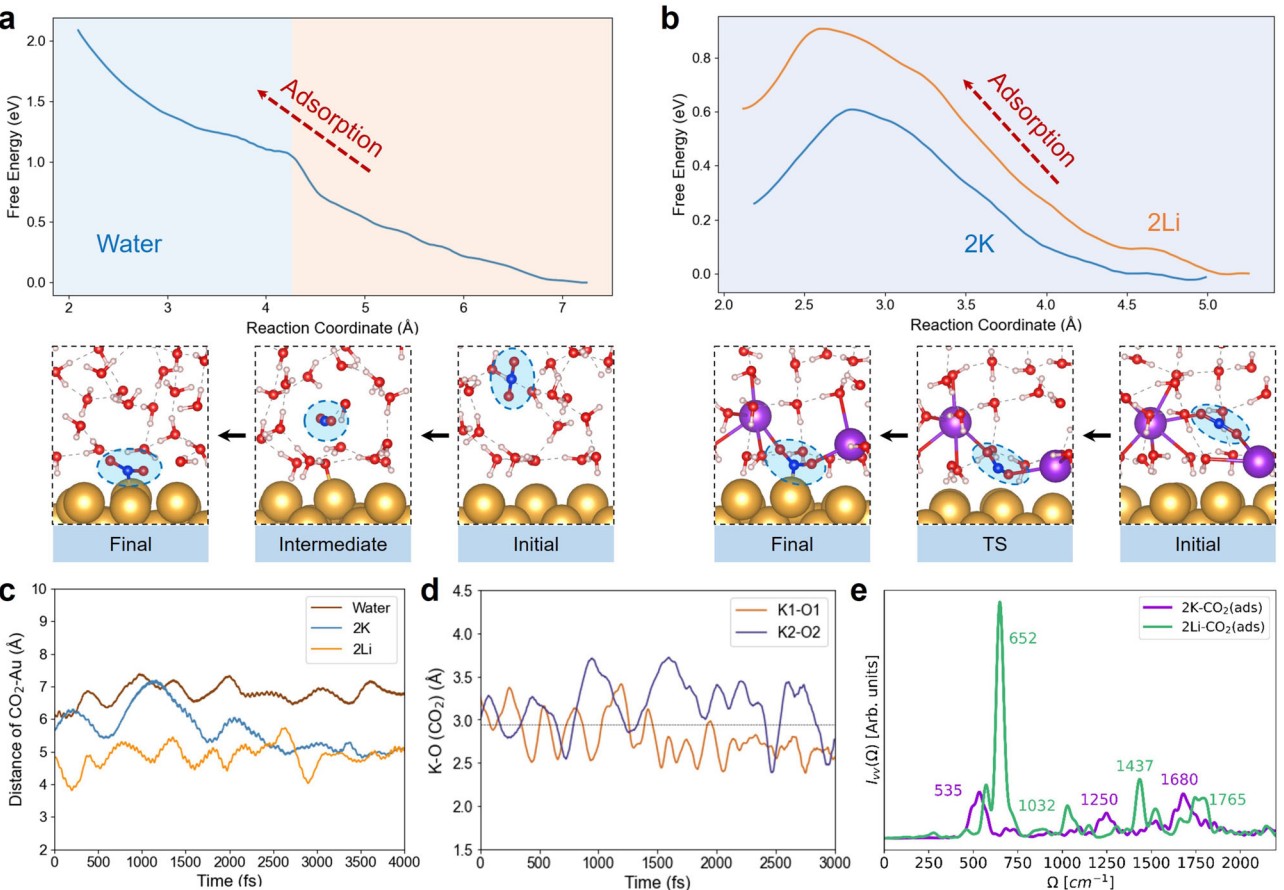

**Fig. 3 | IS-ET reactions via SG-DFT-MD simulations.** Free energy profile (top) and key structures (bottom) of IS-ET at Au-water (**a**) and Au-water-2K (**b**) interfaces, with the free energy curve at Au-water-2Li interfaces inserted for comparison. **c** Distance variations between $CO_2$ and Au surface during 4ps DFT-MD simulations for water and AM$^+$-containing electrolytes. **d** Distance variations of K-O ($CO_2$) during SG-DFT-MD at Au-water-2K interfaces. **e** Simulated vibration spectra of $CO_2$(ads) at Au-water-2K and Au-water-2Li interfaces.

charge analysis indicates that if the $CO_2$ adsorption could occur at Au-water interfaces, it would proceed by partial reduction to $CO_2^{\delta-}(ads)$ with electrons from the Au surface and interfacial water molecules (see Supplementary Fig. S2). However, the IS-ET pathway for $CO_2$ activation cannot proceed without any cations, which is consistent with previous experimental observations that $CO_2$RR cannot take place in cation-deficient electrolytes[12].

Previous simulations have demonstrated that potassium ions (K$^+$) can greatly facilitate the inner-sphere $CO_2$RR including $CO_2$ adsorption and the subsequent reduction on the Au(110) surface[31,32]. Fig. 3b (top, blue curve) shows that the $CO_2$ activation free energy barrier is rather low (0.61 eV) and the adsorption free energy is endergonic by 0.25 eV. The key structures shown in Fig. 3b (bottom) illustrate that $CO_2$ forms a molecular complex with two K$^+$ ions, which are also partially desolvated. At the TS, $CO_2$ is partially reduced to a bent $CO_2^{0.44-}$, which is bound to two interfacial ions closely, and the C-Au distance is 2.74 Å. Compared to K$^+$, Li$^+$ ions can also contribute to the inner-sphere $CO_2$ activation but show a higher energy barrier of 0.91 eV as illustrated in Fig. 3b (top, orange curve) and the adsorption step is endergonic by 0.61 eV. Along the $CO_2$ adsorption at the Au-water-2Li interface (Supplementary Fig. S3), the $CO_2^{\delta-}$ intermediate is less coordinated to Li$^+$ as compared to K$^+$.

## The origin of cation effects

Comparison of the IS-ET kinetics with and without cations (Fig. 3a, b) clearly shows that cations greatly facilitate $CO_2$ adsorption thus enabling the subsequent surface-catalyzed $CO_2$RR steps. To

understand the origin of cation effects on $CO_2$ adsorption, we further analyzed the time-dependent distances between $CO_2$ and the Au surface with and without cations from standard DFT-MD simulations. The results in Fig. 3c indicate that cations attract the neutral $CO_2$ closer to the surface (blue, 5.06 Å for K$^+$; orange, 5.03 Å for Li$^+$) compared to pure water solvent (brown, 6.77 Å). We attribute these differences to the Coulombic interactions between positively charged ions and $CO_2$'s oxygen lone pair with the partial negative charge residing on oxygen (see Supplementary Information Section S5.1 and Fig. S4). During SG-DFT-MD at Au-water-2K interfaces, the K$^+$-O bond distances (Fig. 3d) vary around 2.94 Å, which is close to the bond length range of 2.86-2.94 Å in the ionic $KCO_2$ solid[47]. Furthermore, the $CO_2^{\delta-}$(ads) also carries a large negative charge (~ -0.81 e), and the K$^+$-O bond length and O-C-O angle (Table 1) are very close to those found in crystalline $KCO_2$. These structural fingerprints and charge state indicate the interactions between K$^+$ and $CO_2^{\delta-}$(ads) are very similar to the crystalline phase where the binding can be characterized as an ionic bond or complexation rather than mean-field electrostatics or dipole-field interactions. The electronic localization and charge transfer analysis presented in the Supplementary support this view; there are no signatures of covalent bond formation or charge transfer during the $CO_2$–K complex formation (see Supplementary Information Section S5.2, Fig. S5, and Table S2). Similarly, for Au-water-2Li interfaces, the DFT-MD trajectories of $CO_2$(ads) suggest that both C-O bond lengths and the O-C-O bending angle (Table 1) are close to those in crystalline $LiCO_2$[48], while the short-range coordination between Li$^+$ and $CO_2^{\delta-}$ during SG-DFT-MD is less obvious than that in K$^+$

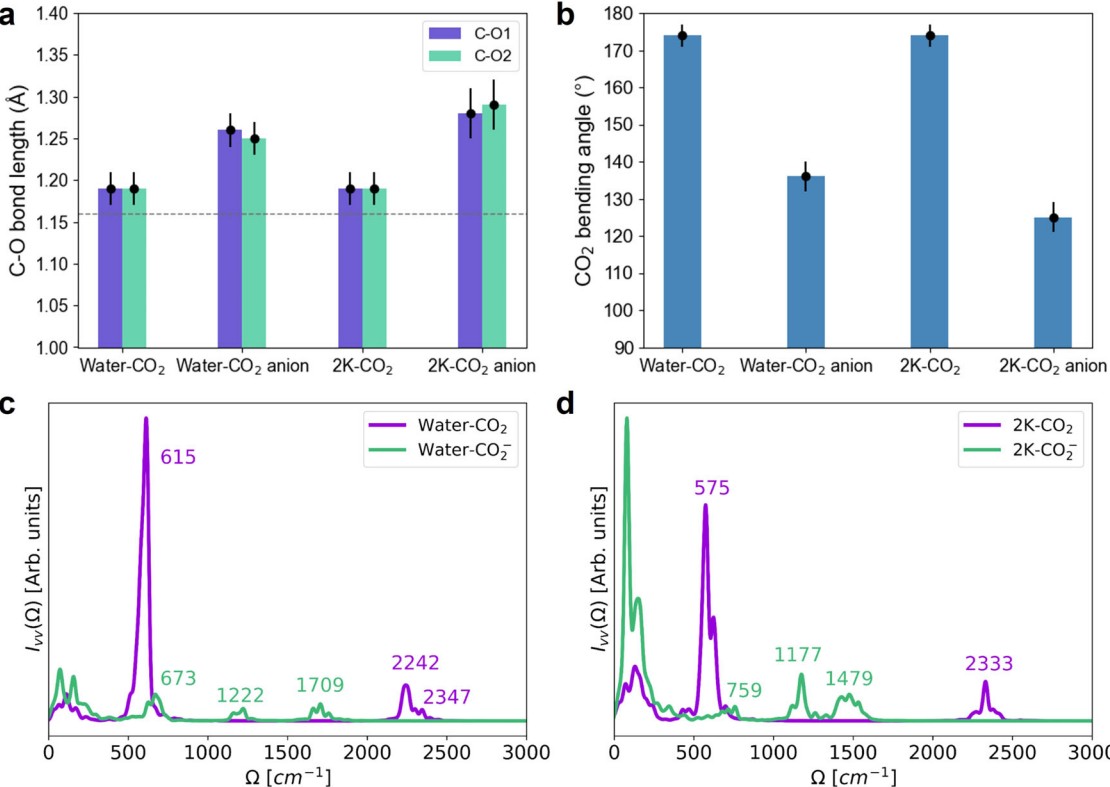

**Fig. 4 | Trajectory analysis of 10ps cDFT-MD simulations.** For $CO_2$ and $CO_2$ anion at Au-water and Au-water-2K interfaces, these analyses include averaged C-O bond length (**a**), bending angle (**b**) with error bars, the simulated vibration spectra of $CO_2$ and $CO_2^-$ for Au-water (**c**) and Au-water-2K (**d**) interfaces. The gray line in (**a**) denotes the C-O bond length of neutral $CO_2$.

(Supplementary Fig. S3b), leading to the higher activation energy for IS-ET. Overall, all the calculated characteristics strongly indicate that alkali metal cations (AM⁺), especially K⁺ ions, promote $CO_2$ activation through short-range Coulombic interactions which lead to direct $CO_2^{\delta-}$–AM⁺ coordination during the IS-ET process.

On the contrary, the OS-ET process exhibits qualitatively different AM⁺-$CO_2$ interactions. In OS-ET, the C-O bond lengths and O-C-O angle are rather independent of the presence of cations (Fig. 4a, b and Table 1). Unlike in the IS-ET simulations (Fig. 3d), there is very minor direct coordination between cations and $CO_2$ or $CO_2^-$ during the cDFT-MD trajectories in both K⁺ and Li⁺ systems (Supplementary Fig. S6). For the neutral $CO_2$, this is expected because cations and $CO_2$ interact mainly through weak van der Waals or induced dipolar interactions. However, for the negative $CO_2^-$, this is rather surprising but can be understood by considering interactions between AM⁺ and $CO_2^-$ as well as their solvation properties. The formation of solvated $CO_2^-$–AM⁺ complexes is thermodynamically favorable[49], but both $CO_2^-$ and cations interact strongly with water and have hydration enthalpies of −1.72, −2.03, and −3.66 eV for $CO_2^-$, K⁺, and Li⁺, respectively[49]. Thus, we expect that $CO_2^-$–AM⁺ complex formation needs substantial activation energy arising from the need to at least partially break the solvation shells around both $CO_2^-$ and AM⁺. Table 1 furthermore shows that in the presence of K⁺ and Li⁺, $CO_2^-$ has a more elongated C-O bond and a smaller O-C-O angle compared to those in pure water. This indicates that the OS-ET step in the presence of cations requires larger geometric changes in the electrolyte liquid compared to pure water. These substantial geometric changes result in high energy gap values and variances, and thereby lead to the high reorganization and reaction energies, respectively. These unfavorable local environments are likely the primary reason for the high reaction energy and sluggish kinetics. We therefore propose that the different solvation properties of $CO_2$ and $CO_2^-$ along with differences in geometric changes are the

underlying cause for the opposite cation effects in IS-ET and OS-ET $CO_2$ activation.

To further shed light on the cation effects in the OS-ET pathway, we compute the vibrational spectra of $CO_2$ and $CO_2^-$ as well as water rotational dynamics in both pure water and cation-containing solutions. As water rotational dynamics are very similar for the two solutions (see Supplementary Fig. S7 and Table S3), changes in water dynamics or rigidity cannot explain the cation dependency of OS-ET reorganization energies. Meanwhile, the analysis of vibration spectra, presented in Fig. 3e, 4c, and d proves to be more fruitful. Table 1 shows that the computed frequencies and mode assignment agree well with NIST data and previous surface-enhanced Raman spectroscopic (SERS) measurements. Most differences can be attributed to slightly different chemical environments. While all frequencies for neutral $CO_2$ are quite similar in both water and cation-containing solutions, the spectra for the $CO_2^-$ deviate. The most notable difference is the change in the C-O stretching modes for water-$CO_2^-$ (1222 and 1709 cm⁻¹), which are shifted to lower frequencies in the presence of K⁺ (1177 and 1479 cm⁻¹) indicating the weakening of the C-O bond. Surprisingly, the O-C-O bending frequency changes from 575 cm⁻¹ for the neutral species to 759 cm⁻¹ for the ionic $CO_2^-$ in the presence of K⁺. This means that K⁺ makes $CO_2^-$ bend faster and more rigid compared to the neutral $CO_2$. Similarly, Li⁺ shows a comparable influence on both $CO_2$ and $CO_2^-$ as compared to K⁺ for bending and stretching modes (Table 1 and Supplementary Fig. S8). Hence, the cation-induced $CO_2^-$ rigidity is another reason for the higher OS-ET reorganization energy in the AM⁺-containing solution.

Figure 4c, d and Table 1 show that the stretching frequencies of $CO_2$ clearly differ from those of $CO_2^-$. In particular, solvated $CO_2$ and $CO_2^-$ can be identified using their different symmetric stretching frequencies, whereas $CO_2^-$ in water and $CO_2^-$–AM⁺ in an electrolyte solution show distinct asymmetric stretching frequencies (Table 1).

However, distinguishing a solvated $CO_2^-$ from an adsorbed $CO_2^{\delta-}$ at Au-water-2K interfaces is very difficult as the stretching frequencies differ less than 50 cm$^{-1}$: the water·$CO_2^-$ system exhibits stretching modes at 1222 and 1709 cm$^{-1}$ (Fig. 4c, Table 1) and 2K-$CO_2^{\delta-}$(ads) stretches at 1250 and 1680 cm$^{-1}$ (Fig. 3e, Table 1). Furthermore, the bending frequencies are difficult to separate from water rotations and translations. Different from 2K-$CO_2^{\delta-}$(ads) (Fig. 3e), 2Li-$CO_2^{\delta-}$(ads) shows two separate peaks in the symmetry stretching (1032 and 1437 cm$^{-1}$), and both the bending and asymmetry stretching are shifted to higher wavenumbers (652 and 1765 cm$^{-1}$, respectively), demonstrating weaker interactions between $CO_2^{\delta-}$(ads) and Li$^+$ compared to K$^+$ in the inner-sphere interfacial region.

Overall, the above frequency analysis demonstrates that solvated $CO_2$ and $CO_2^-$ have distinct vibrational fingerprints in water and cation-containing electrolytes. We found that the C-O bond lengths and O-C-O angle of $CO_2$ do not depend on the presence of cations. $CO_2^-$ is more sensitive to cations (K$^+$ and Li$^+$) which elongate the C-O bond and diminish the O-C-O angle. These findings indicate that OS-ET pathway with cations requires a larger geometric change which leads to a high reorganization energy and slow kinetics. The above results also indicate that spectroscopic identification alone is insufficient to explain the CO$_2$RR mechanism, identify intermediates, or understand cation effects.

## Discussion

We have studied cation effects on outer- and inner-sphere ET thermodynamics and kinetics of the $CO_2$-to-$CO_2^-$ step, which is both the initial and rate-determining step of the CO$_2$RR at Au-water interfaces. We find the OS-ET pathway to be operational only in pure water where $CO_2$ can be reduced to $CO_2^-$ with a surmountable energy barrier of 1.21 eV. The presence of alkali metal cations makes the OS-ET kinetics prohibitively slow as the barrier is very high (2.93 and 4.15 eV for K$^+$ and Li$^+$, respectively) due to high reorganization energy. For the IS-ET process, the opposite trend is true, where K$^+$ cations lead to the low kinetic barrier (0.61 eV) and consequently fast reaction kinetics due to $CO_2^{\delta-}$-cation coordination/complex formation. Without cations, $CO_2$ activation cannot proceed via IS-ET on Au(110) as no stable state for $CO_2^-$(ads) configuration is found.

Besides providing mechanistic insights into cation effects on CO$_2$RR kinetics, we analyzed the cation·$CO_2$ interactions in detail. The charge and bonding analysis results show no signatures of charge transfer or covalent bonding between $CO_2$ and cations (see Supplementary Fig. S5). Instead, we find that cation effects are dominated by short-range, ionic-like coordination bonds mediated through Coulombic interactions (see Supplementary Fig. S4). Our simulations also demonstrate that the complex between partially desolvated K$^+$ and $CO_2^{\delta-}$ persists throughout the $CO_2$ adsorption process in agreement with the cation-coupled ET model[22].

To reinforce the conclusion that short-range interactions between cations and intermediate are behind the cation effects, we also evaluated the role of longe-range electrostatics. Specifically, we address the local electric field modulation due to the cation accumulation in the OHP, which has been proposed as the main mechanism behind cation-enhanced CO$_2$RR[14,16]. To quantify the possible electric field effect, we built a larger $(4 \times 3)$ supercell model with the same number of K$^+$ cations at Au-water interfaces to achieve a weaker local electric field compared to the smaller $(2 \times 3)$ supercell used in all other simulations herein (see Supplementary Fig. S9 and Table S4). We repeated the SG-DFT-MD calculations to obtain the IS-ET potential energy surface for the larger unit cell to study the field and cation-coordination effects. Based on the capacitor model where the interfacial electric field is inversely proportional to the surface area[50,51], the $(4 \times 3)$ supercell possesses half of the averaged electric field strength compared to the smaller simulation cell, but the local cation-coordination between $CO_2$ and K$^+$ is kept unaltered. Additionally, the structure

sampling indicates that the larger supercell shows a greater work function of 3.66($\pm 0.44$) eV and thus a more positive applied potential (-0.37 V$_{RHE}$, reversible hydrogen electrode) than the small supercell (-1.17 V$_{RHE}$)[31]. However, the IS-ET free energy barrier for the $(4 \times 3)$ supercell is calculated to be 0.63 eV, quite close to 0.61 eV in the $(2 \times 3)$ model (see Supplementary Fig. S10), suggesting that ion coordination has a relatively stronger impact on reaction kinetics compared to the electric field and electrode potential. However, it is also important to note that the stronger average electric field at the OHP stabilizes $CO_2^{\delta-}$(ads) more thus enhancing $CO_2$ adsorption. Electric fields can also facilitate subsequent reduction steps since the $CO_2$ bending state, C-O bond lengths, and transferred charge increase under higher electric fields as shown in Supplementary Table S5.

The additional SG-DFT-MD simulation with two K$^+$ cations but without explicit K$^+$–$CO_2$ coordination was carried out in the $(4 \times 3)$ supercell to disentangle short-range coordination contribution from electric field response in cation promotion effects (see Supplementary Fig. S11). The obtained results indicate that no stable $CO_2$ adsorption configuration or TS is found in the weaker electric field without cation coordination as the free energy keeps increasing upon $CO_2$ approaching the Au surface (see Supplementary Fig. S12). This finding is very comparable to our IS-ET results in pure water, and emphasizes the crucial role of direct cation·$CO_2^-$ coordination. Furthermore, our simulations on electric field effects reinforce our conclusion that cations facilitate the initial $CO_2$ activation by the short-range coordination; while stronger local electric fields can stabilize the adsorbed $CO_2^{\delta-}$(ads), the direct interaction between $CO_2$ and AM$^+$ is responsible for the specific cation effect in the first $CO_2$ ET step. These direct cation-reactant interactions modulate the IS- and OS-ET pathways, facilitating IS-ET reaction kinetics and hindering OS-ET of CO$_2$RR.

Our results also provide atomistic insights into the recent work[24] about the role of specific cation adsorption on Au-catalyzed CO$_2$RR. As Fig. 3b shows, a cation spontaneously adsorbs on the surface and coordinates $CO_2$ during the IS-ET pathway. Overall, the computed reaction thermodynamics and kinetics explain the need for cations[12], the role of adsorbed cations[24], and the molecular origins of cation-enhanced CO$_2$RR; the $CO_2$ electroreduction should follow the IS-ET pathway which is facilitated by explicit cation coordination interactions.

We further expect that inner-sphere cation promotion effects can be utilized to enhance CO$_2$RR activity and Faradaic efficiency. Primarily, quasi-specifically adsorbed cations are needed to achieve high activity. This requires a sufficiently high electrolyte concentration and negative surface charge to attract cations to the surface. This is also supported by the recent work categorizing the CO$_2$RR as cation-coupled electron transfer process[22]. On the other hand, if the interfacial cation concentration is too high (e.g., forming the rigid layer), cations may inhibit the $CO_2$ transfer to the surface and an outer-sphere mechanism would be followed. In alkaline HER studies, it has been proposed that the "rigidity" of interfacial water leads to higher reorganization energies and slower HER kinetics. It is possible that a similar effect could be observed in CO$_2$RR, where the lower cation concentration may result in a smaller reorganization energy while maintaining the stabilizing $CO_2^{\delta-}$-cation interaction. The importance of short-range coordinative bonding between $CO_2^{\delta-}$ and cations further suggests that the quasi-specifically adsorbing small but weakly solvated cations would be ideal for promoting the initial steps of CO$_2$RR. Besides facilitating the $CO_2$ reduction, local cation concentration at electrode-electrolyte interfaces can also contribute to hindering the competitive hydrogen evolution[52]. However, it should be noted that addressing the influence of local cation concentrations poses a significant challenge for computational studies because simulating very low concentrations (~0.001M) using DFT-MD is not feasible, and alternative methods should be developed to access such conditions.

In summary, we have used extensive cDFT-MD and SG-DFT-MD simulations to demonstrate that alkali metal cations have a crucial role in modulating the inner- and outer-sphere mechanisms and kinetics of the first $CO_2RR$ step at a Au-wate interface. Through comparison of OS-ET and IS-ET reaction kinetics both with and without cations, we identify that cations facilitate the inner-sphere $CO_2$ activation and electron transfer by forming a $AM^+-CO_2^{\delta-}$ coordination complex. The cation-coordinated $CO_2^{\delta-}$(ads) is the key intermediate and initiates the overall $CO_2RR$ to more reduced products on Au surfaces. In the absence of cations, only outer-sphere ET to $CO_2^-$ (sol) is operational but kinetically hindered. Our detailed analysis of simulation results shows that cations control $CO_2RR$ mainly via short-range, Coulombic interactions rather than long-range, mean-field electrostatics. The computed spectra agree well with previous experimental data allowing us to confirm or interpret previous in situ/*operando* spectro-electrochemical results, but also demonstrate that differentiation between IS-ET and OS-ET pathways cannot be achieved by vibrational spectroscopy alone. Overall, our advanced simulations and detailed analysis clarify how cations control the mechanism, thermodynamics, and kinetics of the initial $CO_2RR$ step.

## Methods

### Model set-up of electrochemical interfaces

We focused on the Au(110) surface which is one of the most active gold facets for $CO_2RR$ with at least 20-fold higher activity compared to other gold surfaces (e.g., Au(100))[53]. The Au(110) slab with seven-atomic layers in a (2 × 3) supercell is constructed with periodic boundary conditions. The box size is $8.32 \times 8.82 \times 40$ Å$^3$ and includes 44 $H_2O$ molecules with a density of ~ 1g·cm$^{-3}$. A vacuum of 12 Å in the $z$ direction is included to prohibit spurious periodic interactions (denoted as Au-water model). Two alkali metal atoms are introduced into the water phase by replacing two $H_2O$ molecules thus constructing the Au-water-2K and Au-water-2Li model systems, to study the cation effect on electron transfer reactions during $CO_2$ activation. A K/Li atom spontaneously donates an electron to the electrode and becomes positively charged while the electrode becomes negatively charged. The overall unit cell is charge-neutral.

In theoretical models, the interfacial concentration of two cations is estimated to be approximately 2.3M within a system containing 42 water molecules. Experimental conditions typically involve cation concentrations ranging from 0.1M to 0.5M[54,55], and cation effects have indeed been observed at cation concentrations as low as 0.001M[12]. Determining the precise interfacial ion concentration is challenging as it depends on factors such as electrode potential and electrolyte properties. Nevertheless, it has been observed that a surface concentration that is 10 to 15 times higher than the experimental bulk concentration provides a reasonable description of interfaces[38]. It is important to note that current DFT simulations are unable to accurately model conditions with very low cation concentrations, such as 0.001M. It should be mentioned that cations and $CO_2/CO_2^-$ do not retain their full bulk solvation shell due to the high surface cation concentration.

### Outer-sphere ET simulations

OS-ET rates are computed using Marcus theory and assuming that the reaction is electronically weakly adiabatic[37]. In this case, the Marcus theory rate can be expressed as

$$k_{Marcus} = \frac{k_B T}{h} \exp\left[-\frac{(\Delta A + \lambda)^2}{4 k_B T \lambda}\right] = \frac{k_B T}{h} \exp\left[-\frac{\Delta A^\ddagger}{k_B T}\right] \quad (1)$$

where $\Delta A, \lambda$, and $\Delta A^\ddagger$ are the reaction Helmholtz free energy, reorganization energy, and OS-ET barrier, respectively. Within the linear response theory, $\Delta A, \lambda$, can be computed from the expectation value ($\langle E \rangle$) and variance ($\sigma$) of the instantaneous diabatic energy gap

($\Delta E(\mathbf{R}(t))$) from constrained DFT-MD (see below) trajectories ($\mathbf{R}(t)$), see Fig. 2. Note that $\lambda$ refers to the total reorganization energy and contains contributions from both the inner-sphere ($CO_2$) and outer-sphere (solvent) reorganization effects.

Because the $CO_2$ and $CO_2^-$ exhibit different solvent responses and solvation properties, an asymmetric version of Marcus theory is used. In this formulation the initial (I, $CO_2$) and final (F, $CO_2^-$) state reorganization energies are different, and two energy gap expectation values are needed:

$$\langle \Delta E \rangle_I = \langle E(CO_2^-, \mathbf{R}(t)) - E(CO_2, \mathbf{R}(t)) \rangle_I \quad (2)$$

$$\langle \Delta E \rangle_F = \langle E(CO_2^-, \mathbf{R}(t)) - E(CO_2, \mathbf{R}(t)) \rangle_F \quad (3)$$

where $\Delta E$ is the instantaneous energy of state I in geometry $\mathbf{R}$ at time $t$. The notation $\langle X \rangle_y$ indicates that the expectation value of $X$ is computed by sampling the state $y$ which for $CO_2$ is obtained from DFT-MD and for $CO_2^-$ from constrained DFT-MD. The variances are defined as[41]

$$\sigma_F^2(\Delta E) = \langle (\Delta E - \langle \Delta E \rangle_F^2) \rangle_F \quad \text{and} \quad \sigma_I^2(\Delta E) = \langle (\Delta E - \langle \Delta E \rangle_I^2) \rangle_I \quad (4)$$

The reorganization and reaction free energies are then computed from the energy gap variances and expectation values as follows[56], and both the reaction energy and reorganization energy depend on the energy gap distributions.

$$\lambda_I = \frac{\sigma_I^2(\Delta E)}{2 k_B T}, \lambda_F = \frac{\sigma_F^2(\Delta E)}{2 k_B T}, \lambda = \frac{1}{2}(\lambda_I + \lambda_F) \quad (5)$$

$$\langle \Delta E_I \rangle = \Delta A + \lambda_F, \langle \Delta E_F \rangle = \Delta A - \lambda_I \quad (6)$$

$$\Delta A_{Marcus}^\ddagger = \frac{(\Delta A + \lambda)^2}{4\lambda} \quad (7)$$

Such computations are based on the linear response form of Marcus theory. $\Delta A_{Marcus}^\ddagger$ arises only when the diabatic energy curves are harmonic along the energy gap coordinate: this condition is met when the energy gap distributions are Gaussian, which is equivalent to the linear response approximation as well as the second cumulant expansion[41,57]. Further discussions of linear and non-linear forms of Marcus theory can be seen in the supplementary information (Supplementary S.1.3).

### DFT simulations

The OS-ET simulations are carried out using the LCAO mode[58] as implemented in GPAW[59–61]. We used a dzp basis for elements[58] and only the Γ-point was used for sampling the reciprocal space. The exchange-correlation effects were modeled using the Perdew-Burke-Ernzerhof (PBE) functional[62] with the Tkatchenko-Scheffler (TS09) dispersion correction[63,64].

### Constrained DFT simulations

The diabatic or charge-localized states are generated using constrained DFT as implemented in the GPAW code[65]. The $CO_2$ and $CO_2^-$ diabatic states are constructed by forcing the $CO_2$ molecule to carry either zero or -1 charge, respectively (Supplementary S.1.4). The other simulation details are the same as those used for the normal DFT calculations in GPAW.

### Molecular dynamics simulations

The GPAW (constrained) DFT-MD simulations are carried out in the canonical ensemble. The temperature is set to 300 K using Langevin

dynamics with a friction parameter of 5 ps$^{-1}$ suitable for water and the time step was set to 1.0 fs.

DFT-MD simulations are performed via Vienna Ab-initio Simulations Package (VASP)[66,67] with the projector augmented wave (PAW) method[68,69]. The Perdew-Burke-Ernzerhof (PBE) functional[62] within the generalized gradient approximation (GGA) framework describes the electron exchange-correlation interactions. The cut-off energy is 400 eV. The k-point mesh grid of $(3 \times 3 \times 1)$ is used. The Methfessel-Paxton smearing is used with a width of 0.2 eV. During dynamic calculations, the bottom four layers of metal are fixed with the rest fully relaxed. The canonical (NVT) ensemble at 298 K is employed with the Nose-Hoover thermostat and the time step is 1.0 fs[70,71]. The zero-damping DFT-D3 method of Grimme[72] is introduced to consider dispersion corrections in all calculations.

### Enhanced sampling and inner-sphere ET simulations

Constrained DFT-MD simulations with the slow-growth (SG) sampling approach[73,74] (denoted as SG-DFT-MD), are performed to evaluate the kinetic barriers of IS-ET during $CO_2$-to-$CO_2^-$ conversion. In this method, the transformations from the initial to the final state can be evaluated along a chosen reaction coordinate, which is defined by a suitable collective variable (CV). For the IS-ET kinetic study, the C-Au distance is chosen as the CV. The transformation step size is controlled to be 0.001 Å for each constrained DFT-MD step. The blue-moon ensemble with SHAKE algorithm as implemented in VASP is adopted to compute the corresponding mean force acting on the CV along the reaction coordinate[75]. The reaction barriers are obtained by integrating the free-energy gradients to compute the free energy profiles based on thermodynamic integrations[75,76]. Further details can be found in the supplementary materials.

### Vibrational spectra and water dynamics

The vibrational spectra and water dynamics are computed from velocity-velocity and dipole-dipole autocorrelation functions, respectively, obtained from the (constrained) DFT-MD simulations. The technical details can be found in the supplementary material.

## Data availability

The representative data and extended datasets that support the findings reported in this study are available in the manuscript and the Supplementary Information. Additional data are available from the corresponding author upon reasonable request. The DFT- and cDFT-optimized geometries and MD trajectories are available in the Zenodo data repository at https://doi.org/10.5281/zenodo.7503348.

## Code availability

The codes for computing energy gaps and simulating vibration spectra have been deposited in the Zenodo data repository at https://doi.org/10.5281/zenodo.7503348.

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

## Acknowledgements

M.M.M. acknowledges the financial support by the Academy of Finland (CompEL project, #338228). The authors acknowledge the computer resources provided by CSC - IT CENTER FOR SCIENCE LTD. H.A.H acknowledges support from the Carlsberg Foundation Young Researcher Fellowship (Grant No. CF19-0304) and Villum Fonden through the project V-sustain (No. 9455).

## Author contributions

X.Q. carried out all MD simulations, analyzed the results, implemented the analysis scripts for the IS-ET simulations, and wrote the initial version of the manuscript. M.M.M. developed the procedure and analysis methods for the OS-ET simulations, implemented the vibrational and rotational dynamics, and performed the bonding analysis. K.H. and H.A.H. supervised the work and contributed to the analysis. All authors contributed to writing the manuscript.

## Competing interests

The authors declare no competing interests.
