## [Peer review file · Nature Communications]

Reviewers' comments:

Reviewer #1 (Remarks to the Author):

I thank the authors for carefully answering all my points. With the revisions, I am sure that the paper will be a great fit to Nature Communications.

Reviewer #2 (Remarks to the Author):

I provided two substantive concerns that lead to the conclusion that this work is not a useful step forward for the field. The first is that the MD simulations are not performed at constant potential (or likely even equivalent potential ranges in situations that are compared) and the second that the MD simulations do not reach sufficient time for results to be significant. Neither of these concerns are improved in this version of the paper, and remain limitations that lead me to the conclusion the technique used is not appropriate for study of the research question. I do not recommend publication of this work, for the same reasons expressed in my original review.

Reviewer #3 (Remarks to the Author):

The authors have responded in detail to the points raised by the four reviewers of the initial submission of this manuscript to Nature Catalysis which I appreciate. Still, most of my main concerns with respect to this manuscript have not been satisfactorily addressed, as I will describe in detail below. Hence I still cannot recommend publication of this manuscript. This is unfortunate as this is in principle a very nice paper by authors that are real experts in the field of the theoretical and computational modeling of electrochemical interfaces. Yet, some of the claims in this paper with respect to the interpretation of the experiments the authors are trying to simulate appear to be not fully substantiated by the model and the results of the simulations.

In my second report, I will use the numbering of my initial report.

1-2) The first two points of the initial report have been appropriately addressed.

3) This point has not been satisfactorily addressed. In response to my remark, the authors have added an explanation about the differences between reorganization and reaction energies, but I did not ask for such an explanation. Rather I have just been wondering why the reaction energy is 0.00 eV. Is it a coincidence that it is basically zero or is there any possible reason for that?

4) response okay

5) I had been asking whether a highly concentrated cation concentration of 2M is realistic. The authors wrote in their response that they also "note that even in joint computational-experimental studies, where the experiments are performed at low ion concentrations and simulations in 10-20 times higher concentrations, the simulations have been able to explain the experimental observations", and then they give three references, the first of which is Ref. 12. In the text, they add a statement: "In the theoretical models, the interfacial concentration of two cations within 42 water molecules is approximately 2.3M. However, in experiments, cation concentrations of 0.1 ~ 0.5M are commonly

used." And in the discussion the authors mention that they provide "a kinetic and mechanistic interpretation to the experiments in Ref. 12", so apparently they particularly address the experiments reported in Ref. 12. Yet, in Ref. 12 one finds the statement: "Experiments were performed in 1 mM M₂SO₄, where M = H, Li, Na, K or Cs." So the main experiments the authors want to explain used a cation concentration that is 100 times smaller than they admit in the paper. Hence this statement in the text needs to be revised.

6) As far as the response of the authors to my point 6 is concerned, I fully admit that it is at the moment impossible to run a simulation at a bulk concentration of 0.1 M, let alone at 0.001 M. Still the issue of the rather high and possibly unrealistic concentrations used in the simulations needs to be appropriately addressed in the discussion of the results.

7) response okay

8) The authors now satisfactorily describe how the K and Li cations are introduced into the system.

9) In their response to my point 9, the authors state that it "appears that the reviewer might have some doubts regarding the cDFT method, which require further clarifications." I would like to emphasize that I do not have doubts regarding the cDFT method. I just wanted the authors to clearly describe how the calculations have been performed so that the reader can understand the implications of adding "ions" to the simulation cell.

The authors have now added Table S1 in the SI in which the Fermi levels of four different situations are listed. I am a little bit confused that the authors give the positions of the Fermi energy, in particular as they do not give the reference for the energy scale. I suppose that the reference is vacuum so that the negative of these Fermi energies would yield the work function. I also wonder why the authors "have estimated the impact of constant potential corrections within the capacitor model.[2]" The authors have used a vacuum region of 12 Å in their simulation cell, so they could have directly determined the work function by monitoring the work function along the runs. Furthermore, apparently the capacitor model in Ref. 2 of the SI is only approximative. As it is written in Ref. 2: "For simple proton transfers, without adsorbates with strong dipoles that reorient dramatically or heavy solvent reorganization, the electrostatic component is purely capacitive, $E_{el} = E_{cap}$." Is this the capacitor model the authors are referring to? Then it might not be fully applicable as solvent reorganization apparently plays a significant role in the system considered here.

When comparing the 2K-CO₂ and the water-CO₂ systems, everything appears to be okay. Upon replacing two water molecules by two K atoms, the surface becomes more negatively charged, which reduces the work function by 0.28 eV and hence the electrode potential by 0.28 V.

Yet, upon replacing two water molecules by two K atoms of the water-CO₂ anion system, the workfunction increases by 0.08 eV. It appears to be odd that the electrode potential increases upon nominally making the electrode more negative. At the same level, the comparison of the water-CO₂ anion system and the water-CO₂ system also yields a surprising result. Here the surface is again supposed to become more negative when the CO₂ anion is replaced by one neutral CO₂, but again the Fermi level becomes lower and the corresponding electrode potential higher. So there seems to be something odd with the Water-CO₂ anion system that needs to be clarified.

10) response okay

11) I like to add another point which I only realized after re-reading the manuscript. In the discussion, the authors write: "the IS-ET free energy barrier for the (4 × 3) supercell is calculated to be 0.63 eV, quite close to 0.61 eV in the (2 × 3) model (see Supplementary Figure S10)". This figure used to be Fig. S6 in the original submission. However, in this figure I cannot see any maximum or barrier, just an free energy curve that is monotonously rising. It might well be that a maximum evolves close to

where the curve ends in Fig. S10, but this is not evidenced by Fig. S10. Hence the authors should weaken their statement.

12) And there is another point that I also only became aware of after completion of the previous report. In the introduction, the authors state that " Complementary density functional theory (DFT) simulations[12] explained this through the stabilization of an adsorbed CO₂ (ads) intermediate by partially desolvated or specifically adsorbed[24] cations with the short-range interaction, in agreement with the cation-coupled electron transfer picture.[22]"

However, this stabilization is apparently not correct. The authors should have a look at Nat. Catal. 5, 977 (2023), doi.org/10.1038/s41929-022-00876-2, by Le and Rahman who showed that the stabilization of the CO₂ intermediate in the calculations reported in Ref. 12 has obviously only be achieved by fixing the distance of the C atoms of CO₂ from the electrode. This reference needs to be considered by the authors.

Reviewer #1:

I thank the authors for carefully answering all my points. With the revisions, I am sure that the paper will be a great fit to Nature Communications.

Response: We thank the reviewer for those solid comments from last round, which helped us to improve our manuscript.

Reviewer #2:

I provided two substantive concerns that lead to the conclusion that this work is not a useful step forward for the field. The first is that the MD simulations are not performed at constant potential (or likely even equivalent potential ranges in situations that are compared) and the second that the MD simulations do not reach sufficient time for results to be significant. Neither of these concerns are improved in this version of the paper, and remain limitations that lead me to the conclusion the technique used is not appropriate for study of the research question. I do not recommend publication of this work, for the same reasons expressed in my original review.

Response: Thanks for the comments. We realized that the reviewer gave us feedback on our manuscript with the same reason as the first round review of *Nature Catalysis* submission. We agree these concerns are important for modelling and simulations. But in our opinions, directly rejecting the paper based on these comments is unfair:

- The cDFT-MD calculations cannot be performed at constant potential – simply not possible.
- The SG-DFT could be done at constant potential now but even this was not possible a year ago.
- We did estimate the electrode potential corrections and found these to be small.
- There's no evidence that the time is too short. The 10ps for the inner-sphere interfacial electrocatalytic reaction simulations is very typical and should be enough for sampling calculations. The Gaussian distribution of the energy provides evidence that the sampling is statistically relevant.

With additional data and discussions, the simulation time of 10ps is not an issue, and the energy corrections towards constant potential condition are too small to affect the overall conclusion and the manuscript. We hope the reviewer could understand this circumstance, and we did try our best with the state-of-the-art simulation methods and theory to elucidate the interfacial mechanism in (CO₂RR) electrocatalysis.

Modifications: Considering the reviewer's comments, we added additional data and discussions about the constant potential and time scale concerns.

Page 7 line 283~286 "It should be mentioned that the OS-ET to form the CO₂⁻ leads to less than 0.3 eV changes in the interface work function, as shown in Supplementary Table S1, which suggests the OS-ET is modeled to occur at a nearly constant potential."

Supplementary Table S1 and S.1.2 "We have estimated the impact of constant potential corrections within the capacitor model.^[2] The resulting energy corrections are quite small as shown in Table S1. For the 2K system, the Fermi level difference between the initial (2K-CO₂) and final states (2K-CO₂ anion) is only 0.28 eV; even if we consider the maximum charge difference of 1e, the energy correction within the capacitor model is only 0.14 eV. For the water system, the Fermi level difference is even smaller (0.08 eV) with a negligible energy correction of 0.04 eV. Comparing the water-CO₂ and 2K-CO₂, the Fermi level difference is also very small, 0.28 eV; water-CO₂ anion and 2K-CO₂ anion show the Fermi level difference of 0.08 eV."

Page 5 line 217~221 “It should be mentioned that longer trajectories and sampling more structures could improve the statistics but already the obtained Gaussian distribution of the energy gap (as required by the microscopic Marcus theory^[39]) tells that the sampling is robust (Figure 2b,e).”

Reviewer #3:

The authors have responded in detail to the points raised by the four reviewers of the initial submission of this manuscript to Nature Catalysis which I appreciate. Still, most of my main concerns with respect to this manuscript have not been satisfactorily addressed, as I will describe in detail below. Hence I still cannot recommend publication of this manuscript. This is unfortunate as this is in principle a very nice paper by authors that are real experts in the field of the theoretical and computational modeling of electrochemical interfaces. Yet, some of the claims in this paper with respect to the interpretation of the experiments the authors are trying to simulate appear to be not fully substantiated by the model and the results of the simulations.

Response: We thank the reviewer for the positive feedback on the quality of our manuscript. By checking the comments in this round, the reviewer’s concern in point 3, 9, 11, and 12, can be addressed with additional data and figure. The disagreement mainly lies in point 5 and 6 on the cation concentration, and we really appreciated that the reviewer also agreed in point 6 that “it is at the moment impossible to run a simulation at a bulk concentration of 0.1 M, let alone at 0.001 M”.

The only remaining concern on the experiments is the concentration used in ref 12 and that we cannot explain these particular experiments – but even the simulations of ref 12 cannot do this because of the high concentrations. Reaching such low concentrations (1 mM) is and will be out of the reach for DFT. Yet, our calculations can explain why and how the cations impact the reaction thermodynamics and kinetics under typical electrolyte conditions such as in e.g. <https://doi.org/10.1021/jacs.1c10171>. Therefore, combining the reviewer’s concerns, the corresponding statements are modified since simulations in this manuscript are not particularly explaining the results in ref 12, but a general understanding towards the crucial cation effects on CO₂RR electrocatalysis.

1-2) The first two points of the initial report have been appropriately addressed.

Response: We thank the reviewer for the nice feedback, and this point was successfully addressed by adding new simulations and analysis of Li cations to demonstrate the cation effects.

3) This point has not been satisfactorily addressed. In response to my remark, the authors have added an explanation about the differences between reorganization and reaction energies, but I did not ask for such an explanation. Rather I have just been wondering why the reaction energy is 0.00 eV. Is it a coincidence that it is basically zero or is there any possible reason for that?

Response: In last round response, we explained the difference between reorganization and reaction energies, since the reviewer gave comments “*I am not questioning the reliability of the results. However, it appears to be striking that the reaction energy is apparently so close to zero, in spite of the fact that the reorganization energy is so large.*” We explain this by adding discussions (Page 5 line 261~262) “It should be noted that the reorganization energy measures the non-equilibrium solvent state, while the reaction energy evaluates the equilibrium solvent state.” We might misunderstand the reviewer’s concern.

Coming back to the current concern “Is it a coincidence that it is basically zero or is there any possible reason for that”, we thank the reviewer for mentioning this concern again. We don’t think it is a coincidence and the possible reason could be explained as follows: Comparing water, K, and Li case,

the reaction energy is 0.00, 2.29, and 3.63 eV, respectively. Since reaction energy calculations are related to the configurations in the sampling (standard deviation and variance), reaction energy difference is also affected by cations and surrounding solvation structures. Without cations, in Au-water system, reaction energy of CO₂-to-CO₂⁻ conversion is 0.00 eV, which makes sense since the CO₂ and CO₂⁻ configurations experience relatively fewer constraints or restrictions compared to the cases involving K and Li cations.

Modifications: Considering the reviewer's concerns, we added additional discussions about the reaction energy comparisons in water, K, and Li systems.

Page 7 line 288~294 "When comparing water, K, and Li case, we observe that the reaction energy of CO₂-to-CO₂⁻ conversion is 0.00, 2.29, and 3.63 eV, respectively, which is affected by cations and surrounding solvation structures. Without cations in the Au-water system, the reaction energy of 0.00 eV makes sense since the CO₂ and CO₂⁻ configurations experience relatively fewer constraints or restrictions compared to the cases involving K and Li cations."

We also keep the last-round modifications about explanations of reorganization energy and reaction energy, which could help readers to better understand the data and related discussions.

Page 5 line 261~262 "It should be noted that the reorganization energy measures the non-equilibrium solvent state, while the reaction energy evaluates the equilibrium solvent state."

4) *response okay*

Response: This point was successfully addressed by demonstrating the fairly robust sampling in cDFT-MD simulations and showing that cDFT approach is reliable for describing the long-range electron transfer and Marcus-type simulations.

5) I had been asking whether a highly concentrated cation concentration of 2M is realistic. The authors wrote in their response that they also "note that even in joint computational-experimental studies, where the experiments are performed at low ion concentrations and simulations in 10-20 times higher concentrations, the simulations have been able to explain the experimental observations", and then they give three references, the first of which is Ref. 12. In the text, they add a statement: "In the theoretical models, the interfacial concentration of two cations within 42 water molecules is approximately 2.3M. However, in experiments, cation concentrations of 0.1 ~ 0.5M are commonly used." And in the discussion the authors mention that they provide "a kinetic and mechanistic interpretation to the experiments in Ref. 12", so apparently they particularly address the experiments reported in Ref. 12. Yet, in Ref. 12 one finds the statement: "Experiments were performed in 1 mM M2SO4, where M = H, Li, Na, K or Cs." So the main experiments the authors want to explain used a cation concentration that is 100 times smaller than they admit in the paper. Hence this statement in the text needs to be revised.

Response: First of all, we really appreciate the patience of the reviewer while reading here. We summarized these concerns as follows: (1) in the first round of review, the reviewer thinks the cation concentration in simulations is too high; (2) in this round, the reviewer mentioned if our simulations are used to particularly explain the experiments in ref 12, the statement needs to be revised.

As response to (1), the cation concentration issue was explicitly addressed by discussing the possibility of local cation accumulations at interfaces as compared to experiments. We added additional discussion in Page 14 line 642~651, and we thank the reviewer for agreeing with this point (*Point 6*). Currently it is also not possible to simulate such a low concentration by DFT.

For the second point (2), our simulations are not particularly used to explain the results in ref 12 with a 1mM concentration, and the cation effect exists in a broad range of experiments with cation

concentration (1mM ~ 0.5M). We agree with the reviewer that the statement needs to be modified, giving our discussions more clearly.

Modifications: We modified the statement to indicate that our simulation is not particularly to explain the ref 12. Page 11 line 504~505 “Besides providing mechanistic insights into cation effects on CO₂RR kinetics, we also analyzed the cation-CO₂ interactions in detail.”

6) As far as the response of the authors to my point 6 is concerned, I fully admit that is at the moment impossible to run a simulation at a bulk concentration of 0.1 M, let alone at 0.001 M. Still the issue of the rather high and possibly unrealistic concentrations used in the simulations needs to be appropriately addressed in the discussion of the results.

Response: We appreciate that the reviewer agrees it is not possible to simulate the exact experimental condition. We thank the reviewer for the understanding. We further addressed the high cation concentration concern in the result discussions.

Modifications: We revised previous discussions about the cation concentration and added discussions about the current limitation in DFT for simulating the experiment condition with low cation concentrations.

Page 14 line 642~651 “In theoretical models, the interfacial concentration of two cations is estimated to be approximately 2.3M within a system containing 42 water molecules. Experimental conditions typically involve cation concentrations ranging from 0.1M to 0.5M. However, determining the precise interfacial ion concentration is challenging as it depends on factors such as electrode potential and electrolyte properties. Nevertheless, it has been observed that a surface concentration that is 10 to 15 times higher than the experimental bulk concentration provides a reasonable description of interfaces. It is important to note that current DFT simulations are unable to accurately model conditions with very low cation concentrations, such as 0.001M.”

7) response okay

Response: This point was explicitly addressed by describing the K-O (CO₂) coordination and solvation shells of cations and CO₂/CO₂⁻.

8) The authors now satisfactorily describe how the K and Li cations are introduced into the system.

Response: We addressed this point by adding explicit discussions of model set-ups.

9) In their response to my point 9, the authors state that it "appears that the reviewer might have some doubts regarding the cDFT method, which require further clarifications." I would like to emphasize that I do not have doubts regarding the cDFT method. I just wanted the authors to clearly describe how the calculations have been performed so that the reader can understand the implications of adding "ions" to the simulation cell.

Response: We apologize if we made sloppy writing by saying “the reviewer might have doubts regarding.....”. In the revised manuscript, the calculation details and model descriptions of adding “ions” were explicitly discussed, which are added into the manuscript.

Modifications: Page 16 line 696~698 “The CO₂ and CO₂⁻ diabatic states are constructed by forcing the CO₂ molecule to carry either zero or -1 charge, respectively (Supplementary S.1.3).”

Supplementary S.1.3, we added method descriptions of constrained DFT, electron transfer, and Marcus theory (Supplementary Page 3 line 93~119).

The authors have now added Table S1 in the SI in which the Fermi levels of four different situations are listed. I am a little bit confused that the authors give the positions of the Fermi energy, in particular as they do not give the reference for the energy scale. I suppose that the reference is vacuum so that the negative of these Fermi energies would yield the work function. I also wonder why the authors "have estimated the impact of constant potential corrections within the capacitor model.[2]" The authors have used a vacuum region of 12 Å in their simulation cell, so they could have directly determined the work function by monitoring the work function along the runs. Furthermore, apparently the capacitor model in Ref. 2 of the SI is only approximative. As it is written in Ref. 2: "For simple proton transfers, without adsorbates with strong dipoles that reorient dramatically or heavy solvent reorganization, the electrostatic component is purely capacitive, $E_{el} = E_{cap}$." Is this the capacitor model the authors are referring to? Then it might not be fully applicable as solvent reorganization apparently plays a significant role in the system considered here.

When comparing the 2K-CO₂ and the water-CO₂ systems, everything appears to be okay. Upon replacing two water molecules by two K atoms, the surface becomes more negatively charge, which reduces the work function by 0.28 eV and hence the electrode potential by 0.28 V. Yet, upon replacing two water molecules by two K atoms of the water-CO₂ anion system, the workfunction increases by 0.08 eV. It appears to be odd that the electrode potential increases upon nominally making the electrode more negative. At the same level, the comparison of the water-CO₂ anion system and the water-CO₂ system also yields a surprising result. Here the surface is again supposed to be become more negative when the CO₂ anion is replaced by one neutral CO₂, but again the Fermi level becomes lower and the corresponding electrode potential higher. So there seems to be something odd with the Water-CO₂ anion system that needs to be clarified.

Response: Again, we appreciate the patience of the reviewer. There are two main concerns here, which are addressed as follows:

Firstly, the reference is vacuum and work function can be yielded from Fermi energy. Fermi energies are also monitored along the runs. Yes, we agree that the capacitor correction is an approximation and we have carefully analyzed the conditions when it breaks down (<https://doi.org/10.1063/5.0138197>). Shortly, these approaches cannot be applied to situations where the adsorption, transition state, or electrolyte configuration change as a function of the electrode potential (from the conclusions of the above paper): this only considers potential-dependency of the geometries and e.g. solvent reorganization. Furthermore, the solvent reorganization corrections in ref 2 correspond to potential-dependent changes in the reorganization – not solvent reorganization due to the reaction. Because we do not consider different potentials, also the reorganization argument is not valid. While the capacitance corrections is not fully accurate, it gives the capacitive correct, i.e. the 2nd order Taylor expansion correction for a *constant* capacitance. Because capacitance mostly depends on the surface and electrolyte, the capacitor model does provide a reasonable guess on the magnitude of the error coming from the constant charge treatment provided that the differential capacitance does not change during the reaction.

Finally, for the charges and work function changes (minor), maybe the description is not very clear, but we don't see anything weird here. Adding cations to the solution with a neutral CO₂ should make the surface more negative as the reviewer writes and this is what we see. In the water case, the Fermi-level is changed only by 0.08 with a variance of 0.25, meaning that it is basically the same. It is also possible that the Fermi-level in the water-CO₂⁻ system is determined by CO₂⁻ rather than the surface.

Modifications: Based on the reviewer's comments, additional discussions are added in the supplementary information to describe the Fermi level changes.

Supplementary S.1.2 "Adding cations to the solution with neutral CO₂ should make the surface more negative, which is observed. In the water case, the Fermi level is changed only by 0.08 with a variance of 0.25 comparing the water-CO₂ and water-CO₂ anion systems, meaning that it is basically the same. It is also possible that the Fermi level in the water-CO₂⁻ system is determined by CO₂⁻ rather than the surface."

10) response okay

Response: We addressed the reviewer's concern via discussing the particular highlights in our current study, focusing on the long-standing question of outer-sphere and inner-sphere CO₂ reduction mechanism. As far as we know, this is the very first time to explicitly simulate the outer-sphere CO₂ reduction pathway combining cDFT-MD and Marcus theory.

11) I like to add another point which I only realized after re-reading the manuscript. In the discussion, the authors write: "the IS-ET free energy barrier for the (4 × 3) supercell is calculated to be 0.63 eV, quite close to 0.61 eV in the (2 × 3) model (see Supplementary Figure S10)". This figure used to be Fig. S6 in the original submission. However, in this figure I cannot see any maximum or barrier, just an free energy curve that is monotonously rising. It might well be that a maximum evolves close to where the curve ends in Fig. S10, but this is not evidenced by Fig. S10. Hence the authors should weaken their statement.

Response: We thank the reviewer for pointing this out. We agree with the reviewer that a maximum evolves quite close to the ending point of the curve. The free energy curve shown in Figure S10 is obtained by thermodynamic integrations with dynamic average, and we also showed the original data without dynamic average and marked this maximum (yellow circle) in the figure below. The additional discussions are added into the revised manuscript and supplementary information.

Fig. R1. The original free energy profile of CO₂ adsorption in the (4 × 3) model with cation coordination.

Modifications: Considering the reviewer's concerns, we further clarified the maximum evolves quite close to the ending point of the curve in Fig. S10. The statement is also modified in the revised manuscript.

Page 12 line 550~557 “However, the IS-ET free energy barrier for the (4 × 3) supercell is calculated to be 0.63 eV, quite close to 0.61 eV in the (2 × 3) model (see Supplementary Figure S10), suggesting that ion coordination has a relatively stronger impact on reaction kinetics compared to the electric field and electrode potential. However, it is also important to note that the stronger average electric field at the OHP stabilizes CO₂^{δ-} (ads) more thus enhancing CO₂ adsorption.”

12) And there is another point that I also only became aware of after completion of the previous report. In the introduction, the authors state that "Complementary density functional theory (DFT) simulations [12] explained this through the stabilization of an adsorbed CO₂(ads) intermediate by partially desolvated or specifically adsorbed[24] cations with the short-range interaction, in agreement with the cation-coupled electron transfer picture.[22]"

However, this stabilization is apparently not correct. The authors should have a look at Nat. Catal. 5, 977 (2022), doi.org/10.1038/s41929-022-00876-2, by Le and Rahman who showed that the stabilization of the CO₂ intermediate in the calculations reported in Ref. 12 has obviously only be achieved by fixing the distance of the C atoms of CO₂ from the electrode. This reference needs to be considered by the authors.

Response: First of all, we thank the reviewer for mentioning these disputes regarding the cation stabilization effect. The paper, *Nat. Catal. 5, 977 (2022), doi.org/10.1038/s41929-022-00876-2*, by Le and Rahman does show that the calculations in ref 12 are questionable. However, the follow-up response to this comment paper, *Nat. Catal. 5, 979 (2022), <https://www.nature.com/articles/s41929-022-00877-1>*, shows that the main mechanistic conclusions of ref 12 hold even if the artificial constraint is removed. We’re simply pointing out that our results agree with the scheme proposed in ref.12 as well as ref 22.

We do agree with the reviewer that all of these very recent disputes should be included in the introduction in the revised manuscript.

Modifications: The recent disputes and related references are included into the introduction.

Page 3 line 103~109 “Complementary density functional theory (DFT) simulations [12] explained this through the stabilization of an adsorbed CO₂(ads) intermediate by partially desolvated or specifically adsorbed [24] cations with the short-range interaction, in agreement with the cation-coupled electron transfer picture. [22] Although the calculations in Ref. 12 were questioned by Le and Rahman [28], the main mechanistic conclusions still hold even if the artificial constraint of fixing the distance of the C (CO₂) from the electrode is removed.[29]”

REVIEWER COMMENTS

Reviewer #3 (Remarks to the Author):

The authors have again worked hard to further improve the manuscript.

With respect to point 4, they now give a convincing explanation. Also with respect to my points 9, 11 and 12, I am satisfied with the response of the authors

My main criticism (points 5-6) was still related to the much higher cation concentration used in the calculations than for example in the experiments reported in Ref. 12. However, it is clear to me, as I already admitted in my previous report, that such low cation concentrations as employed in Ref. 12 cannot be realized in first-principles atomistic studies. It might also be true that "experimental conditions typically involve cation concentration ranging from 0.1M to 0.5M", as the authors now added to the text in the Methods section. However, I am not satisfied with the added sentence of the authors: "It is important to note that current DFT simulations are unable to accurately model conditions with very low cation concentrations, such as 0.001M." This sounds as if 0.001M are just model conditions but not realized in experiment. Furthermore, the non-initiated reader will not understand why this statement is placed at this position because no reference is made to any experiment. And it still remains an important fact that surprisingly strong cation effects are observed at rather low concentrations. Now I realize that cation effects in electrocatalysis are a very hot topic at the moment, and that the authors provide a substantial contribution to the discussion about this topic. Still, it should be good scientific practice not to somewhat "hide" important experimental findings. Hence I could now fully support publication of this nice work in Nature Communications once an additional rather minor change has been made to the text, namely that the additional sentence in line 644 of the manuscript is extended to something such as:

"Experimental conditions typically involve cation concentrations ranging from 0.1M to 0.5M [56, 57], but cation effects have in fact been observed at cation concentrations as low as 0.001M [12]."

I would even more appreciate if such a statement is not only made in the Methods section after the Conclusions, but already in the presentation and discussion of the results in the main text. This would even more emphasize that the authors are interested in a thorough discussion of their simulations and results, but I would leave it to the authors to follow this well-meant recommendation.

Reviewer #4 (Remarks to the Author):

In this paper, the authors use DFT-MD simulations to model mechanisms of CO₂ activation over Au(110) single-crystal electrodes. Their study probes the effects of cations on the kinetics of inner- and outer-sphere CO₂ reduction in the initial activation step. The methods are state-of-the-art and this is an excellent demonstration of the ability of charge constraints to derive free energy curves that can enable diabatic representations of electrochemical reactions. Overall, I believe the methods presented are impactful and provide a fresh approach to modeling inner- and outer-sphere reactions in heterogeneous electrocatalysis. However, I do have some concerns and further clarifications are needed prior to publication. My comments are listed below:

- The Marcus theory schematic in Fig 2a,d is confusing because it is not obvious from the label that this is an energy difference for a given diabatic state. It looks like it is drawn between the blue and red curves.
- The authors use "reorganization energy" and "solvent reorganization energy" interchangeably. But if I understand correctly, what they are computing is a total reorganization energy (both inner/solute and outer/solvent included) and no attempt is made to deconvolute these two effects. Please clarify.
- I understand that the 10 ps OSET CDFT-DFT-MD simulations should be sufficient to sample the

energy gap, given the Gaussian distributions in all models. The authors should clarify how the energy gap is calculated with CDFT, and how often. Is it calculated at every time step?

- I think that the authors have been clear about changes in Fermi energy between the models, even though some of these results are surprising (e.g., the work function decreases after ET to CO₂ in the water-Au model).
- It is very surprising that the reaction free energies are so large (> 2 eV!) for the AM+-including models. The authors do not attempt to explain why these reaction energies are so large and instead focus on discussion of the reorganization energies/barriers. But if the reaction energy is so large, who cares about the kinetics? There could be zero overbarrier and the reaction still will not happen. I am more interested in hearing the author's thoughts about why this reaction is so thermodynamically unfavorable in the presence of ions. What about the CO₂⁻ diabatic state makes it so unstable relative to the Au-water model? This may or may not be related to my next comment.
- One major difference between the OSET and ISET simulations has to do with the coordination between CO₂/CO₂⁻ and the anions. The specific short-range Coulombic interactions (as indicated by the K-O distances in Fig 3d, for example) lower the free energy along the ISET reaction coordinate. However, the relative positions of the K⁺ and CO₂ shown in Fig 2f suggest that there are no specific interactions between the ions and CO₂. I understand that these are dynamic simulations, but analogous information to that in Fig 3d for the OSET simulations could help clarify.
- One minor related point, page 6. It is mentioned that the reorganization energy is much higher than for Au-water, and that "Consequently, the reaction energy is highly endothermic." I do not believe that the endothermic reaction energy is a consequence of a high reorganization energy. Reaction energies correspond to energy differences between equilibrated reactant and product states and should be agnostic to any information about nonequilibrium solvation.
- In the authors' response to Reviewer #3 (point 9) they have suggested that "it is possible that the Fermi level in the water-CO₂ system is determined by CO₂⁻ rather than the surface." This speculation could be further supported through a projected DOS analysis.

REVIEWER COMMENTS

Reviewer #3 (Remarks to the Author):

The authors have again worked hard to further improve the manuscript.

With respect to point 4, they now give a convincing explanation. Also with respect to my points 9, 11 and 12, I am satisfied with the response of the authors.

Comment 1: My main criticism (points 5-6) was still related to the much higher cation concentration used in the calculations than for example in the experiments reported in Ref. 12. However, it is clear to me, as I already admitted in my previous report, that such low cation concentrations as employed in Ref. 12 cannot be realized in first-principles atomistic studies. It might also be true that "experimental conditions typically involve cation concentration ranging from 0.1M to 0.5M", as the authors now added to the text in the Methods section. However, I am not satisfied with the added sentence of the authors: "It is important to note that current DFT simulations are unable to accurately model conditions with very low cation concentrations, such as 0.001M." This sounds as if 0.001M are just model conditions but not realized in experiment. Furthermore, the non-initiated reader will not understand why this statement is placed at this position because no reference is made to any experiment. And it still remains an important fact that surprisingly strong cation effects are observed at rather low concentrations. Now I realize that cation effects in electrocatalysis are a very hot topic at the moment, and that the authors provide a substantial contribution to the discussion about this topic. Still, it should be good scientific practice not to somewhat "hide" important experimental findings. Hence I could now fully support publication of this nice work in Nature Communications once an additional rather minor change has been made to the text, namely that the additional sentence in line 644 of the manuscript is extended to something such as:

"Experimental conditions typically involve cation concentrations ranging from 0.1M to 0.5M [56, 57], but cation effects have in fact been observed at cation concentrations as low as 0.001M [12]."

I would even more appreciate if such a statement is not only made in the Methods section after the Conclusions, but already in the presentation and discussion of the results in the main text. This would even more emphasize that the authors are interested in a thorough discussion of their simulations and results, but I would leave it to the authors to follow this well-meant recommendation.

Response 1: We completely agree that addressing the matter of simulating (very) low cation concentrations deserves attention. In the revised manuscript, we have brought this issue to the forefront and discussed it as follows.

Modifications:

At the beginning of the Results section (Page 4, line 175~181):

"The interfacial cation concentration used is ~2.3 M, which would correspond to a bulk concentration of roughly 0.1M-0.5 M, as predicted by simulations of cation accumulation at the electrode interface under CO₂RR-relevant conditions.[39] While the cation concentration

of 0.1M-0.5 M is typical for CO₂RR, it has been shown that even cation concentrations as low as 0.001M can impact CO₂RR activity.[12] However, current DFT simulations are unable to model such low concentrations.”

At the end of discussion (Page 14, line 620~624):

“However, it should be noted that addressing the influence of local cation concentrations poses a significant challenge for computational studies because simulating very low concentrations (~ 0.001M) using DFT-MD is not feasible, and alternative methods should be developed to access such conditions.”

Methods section (Page 15, line 666~668):

"Experimental conditions typically involve cation concentrations ranging from 0.1M to 0.5M [57, 58], and cation effects have indeed been observed at cation concentrations as low as 0.001M [12]."

Reviewer #4 (Remarks to the Author):

In this paper, the authors use DFT-MD simulations to model mechanisms of CO₂ activation over Au(110) single-crystal electrodes. Their study probes the effects of cations on the kinetics of inner- and outer-sphere CO₂ reduction in the initial activation step. The methods are state-of-the-art and this is an excellent demonstration of the ability of charge constraints to derive free energy curves that can enable diabatic representations of electrochemical reactions. Overall, I believe the methods presented are impactful and provide a fresh approach to modeling inner- and outer-sphere reactions in heterogeneous electrocatalysis. However, I do have some concerns and further clarifications are needed prior to publication. My comments are listed below:

Response: We thank the reviewer for these encouraging comments. We have explicitly addressed the following concerns by point-to-point below. Furthermore, based on the reviewer’s comments, we have added further discussion on reorganization energy, energy gap calculations, Fermi energies, and reaction free energies, which has led to improvements in our manuscript.

Comment 1. The Marcus theory schematic in Fig 2a,d is confusing because it is not obvious from the label that this is an energy difference for a given diabatic state. It looks like it is drawn between the blue and red curves.

Response 1: Thank you for your comment. Both the ΔA and λ terms in Figure 2a,d represent energetic differences. To emphasize this, we have included the ordinates (y-axes) in Figure 2a,d. Additionally, we have added arrows to better highlight the energetic terms ΔA and λ .

Modifications: The modified Figure 2a and 2d are shown below and corresponding changes are made in Figure 2 within the manuscript.

Figure R1. Schematic view of the Marcus theory in Au-water (a) and Au-water-2K (d).

Comment 2. The authors use “reorganization energy” and “solvent reorganization energy” interchangeably. But if I understand correctly, what they are computing is a total reorganization energy (both inner/solute and outer/solvent included) and no attempt is made to deconvolute these two effects. Please clarify.

Response 2: The reviewer is correct: we are only computing the total reorganization energy and we are not attempting to separate the inner- and outer-contributions as this is very difficult with the explicit solvent model used here.

Modifications:

To acknowledge this issue, we have removed the term “solvent reorganization energy” from the manuscript and refer only to “reorganization energy”. We have also added the following sentence in the Methods section (Page 15, line 688~690):

“Note that λ refers to the total reorganization energy and contains contributions from both the inner-sphere (CO₂) and outer-sphere (solvent) reorganization effects.”

Comment 3. I understand that the 10 ps OSET CDFT-DFT-MD simulations should be sufficient to sample the energy gap, given the Gaussian distributions in all models. The authors should clarify how the energy gap is calculated with CDFT, and how often. Is it calculated at every time step?

Response 3: We appreciate the reviewer for raising this point. In relation to the energy gap sampling, the energy gap is determined by calculating the instantaneous energy difference between the CO₂ and CO₂⁻ states across various geometries and solvent environments. This energy gap is calculated every 50 fs. We incorporated these details into the revised manuscript.

Modifications (Page 5, line 219~221):

“The energy gap is determined by calculating the instantaneous energy difference between the CO₂ and CO₂⁻ states across different geometries and solvent environments with calculations performed every 50 fs.”

Comment 4. I think that the authors have been clear about changes in Fermi energy between the models, even though some of these results are surprising (e.g., the work function decreases after ET to CO₂ in the water-Au model).

Response 4: We acknowledge that some of these results are surprising, such as the work function decreases after ET to CO₂ in the water-Au model. As shown in Table S1, the transition from the water-CO₂ to the water-CO₂⁻ system results in the Fermi level change of only 0.08eV with a variance of 0.25eV. Consequently, the Fermi-level remains essentially unchanged. Nevertheless, it is possible that the Fermi level in the water-CO₂⁻ system is primarily determined by CO₂⁻ rather than the surface in the case of the water-CO₂ system. This hypothesis is supported by PDOS analysis as recommended by the reviewer (see Comment 8 and Response 8).

However, elucidating *why* the Fermi level behaves in this manner is difficult within complex systems. Pinpointing which atoms or molecules determine the Fermi-level or work function in a dynamic system is also a complicated task. Moreover, it raises a more general question of whether the Fermi level of the *simulation cell* as opposed to that of the electrode is the correct quantity to define the electrode potential. Our recent work shows that, for outer-sphere couples, the Fermi level does not correctly characterize the electrode potential (<https://chemrxiv.org/engage/chemrxiv/article-details/63fe1546937392db3d34fa4a>).

However, these questions are beyond the scope of the current work. Therefore, we analyze only the Fermi level changes and the PDOS in response to comments 8, acknowledging their somewhat unexpected behavior, which we are unable to fully explain in this work.

Comment 5. It is very surprising that the reaction free energies are so large (> 2 eV!) for the AM⁺-including models. The authors do not attempt to explain why these reaction energies are so large and instead focus on discussion of the reorganization energies/barriers. But if the reaction energy is so large, who cares about the kinetics? There could be zero overbarrier and the reaction still will not happen. I am more interested in hearing the author's thoughts about why this reaction is so thermodynamically unfavorable in the presence of ions. What about the CO₂⁻ diabatic state makes it so unstable relative to the Au-water model? This may or may not be related to my next comment.

Response 5: We appreciate the constructive comments from the reviewer. Unfortunately, we can only speculate about the origin of the high reaction energy. We see two options: either physical/chemical factors or computational/theoretical issues.

As outlined in the methods section, within the linear response Marcus theory, the reaction energy is determined by the energy gap expectation value. Apart from generating the diabatic states using cDFT, this is the sole assumption in our model and in computing the reaction energy. It is important to note that there is no universally accepted method for defining a diabatic state. The cDFT method falls within a category of empirical valence bond methods, where the definition of diabatic states relies on chemical intuition. However, we consider the cDFT diabatic description of CO₂⁻ to be reasonably accurate, as evidenced by the computed vibrational frequencies of CO₂⁻ closely matching experimental values (Table 1). Additionally, it is possible that the linear response Marcus theory may be too simplistic for modeling the CO₂RR OS-ET reaction, and that non-linear effects should also be considered (<https://doi.org/10.1021/ja2069104>).

Instead of originating solely from the well-established cDFT method and Marcus theory, the large reaction energy may also arise from the physical/chemical processes as well as

interactions. As the reviewer notes in Comment 6, the coordination of $\text{CO}_2/\text{CO}_2^-$ with cations differs significantly between the OS-ET and IS-ET simulations:

(1) During IS-ET simulations, the coordination of $\text{CO}_2/\text{CO}_2^-$ with cations due to short-range (electrostatic) interactions is readily observed.

(2) However, in OS-ET simulations, both CO_2 and CO_2^- exhibit much weaker interactions with cations compared to IS-ET (see Figure in Comment 6).

(3) When we compare the OS-ET reactions in the Au-water and Au-water-cations systems, the presence of cations results in larger geometric changes when converting CO_2 to CO_2^- or *vice versa*. These substantial geometric changes are also evident in the geometry and vibrational wavenumber analysis (Table 1), when comparing the Water- CO_2 , Water- CO_2^- , 2K- CO_2 , 2K- CO_2^- , 2Li- CO_2 , and 2Li- CO_2^- using NIST data as references. Compared to K, the Li cations necessitate even greater geometric changes leading to a higher reaction energy.

In summary, the $\text{CO}_2/\text{CO}_2^-$ equilibrium geometries used in calculating the energy gap (which ultimately determines the reaction energy) are highly unfavorable in systems containing cations compared to Au-water model. These unfavorable local environments are the most likely the primary reason behind the high reaction energy.

Modifications: We have added further discussion to explain the possible reason for the high reaction energy in the OS-ET simulations with Au-water-cations systems, mainly from the physical/chemical aspect (Page 10, line 447~453).

“This indicates that the OS-ET step in the presence of cations requires larger geometric changes in the electrolyte liquid compared to pure water. These substantial geometric changes result in high energy gap values and variances, and thereby lead to the high reorganization and reaction energies, respectively. These unfavorable local environments are likely the primary reason for the high reaction energy and sluggish kinetics.”

Comment 6. One major difference between the OSET and ISET simulations has to do with the coordination between $\text{CO}_2/\text{CO}_2^-$ and the cations. The specific short-range Coulombic interactions (as indicated by the K–O distances in Fig 3d, for example) lower the free energy along the ISET reaction coordinate. However, the relative positions of the K^+ and CO_2 shown in Fig 2f suggest that there are no specific interactions between the ions and CO_2 . I understand that these are dynamic simulations, but analogous information to that in Fig 3d for the OSET simulations could help clarify.

Response 6: We thank the reviewer’s comment. As illustrated in Fig. 2f, there are no specific interactions between ions and CO_2 in the OS-ET simulations for the Au-water-2K system. We fully concur that conducting a similar analysis to that in IS-ET dynamic simulations could assist in elucidating the primary difference between the OS-ET and IS-ET simulations. This analysis has been included as Figure S6 in the supporting information, labeled as Figure R2 below.

As depicted in Figure R2, further examination of K-O (CO_2) and K-O (CO_2^-) distance variations during the OS-ET simulations reveals that short-range interactions are occasionally observed but are not as stable or as prominent as those observed in the IS-ET simulations.

Figure R2. The distance analysis during 10 ps OS-ET simulations at Au-water-2K interfaces including K-O (CO₂) (a) and K-O (CO₂ anion) (b).

Modifications: We have included the above figure in the supporting information and addressed the main difference between IS-ET and OS-ET in the manuscript (Page 9, line 406~409).

“Unlike in the IS-ET simulations (Figure 3d), there is very minor direct coordination between cations and CO₂ or CO₂⁻ during the cDFT-MD trajectories in both K⁺ and Li⁺ systems (Figure S6).”

Comment 7. One minor related point, page 6. It is mentioned that the reorganization energy is much higher than for Au-water, and that “Consequently, the reaction energy is highly endothermic.” I do not believe that the endothermic reaction energy is a consequence of a high reorganization energy. Reaction energies correspond to energy differences between equilibrated reactant and product states and should be agnostic to any information about nonequilibrium solvation.

Response 7: Thank you for the comment. We agree that our discussion on the relation between the reaction energy and the reorganization was too sloppy. Therefore, based on Comment 5 (please see our response there), here we simply modify the expression as follows (Page 7, line 285~286):

Modifications: “Additionally, the reaction energy is highly endothermic being 2.29 eV, which makes the reaction very unfeasible.”

Comment 8. In the authors’ response to Reviewer #3 (point 9) they have suggested that “it is possible that the Fermi level in the water-CO₂⁻ system is determined by CO₂⁻ rather than the surface.” This speculation could be further supported through a projected DOS analysis.

Response 8: Thank you for this valuable suggestion. However, performing the PDOS analysis for the entire trajectory is overly complicated and computationally demanding. As a compromise, we have performed the PDOS analysis at different points of the OS-ET calculation for the CO₂⁻ and the CO₂ species. As shown in Figure R3a, in the water-CO₂⁻ system, there is a clear overlap between CO₂⁻ PDOS and total DOS near to the Fermi level ($E-E_f = 0$ eV), marked by the yellow color. This overlap shows that CO₂⁻ clearly contributes to the Fermi level in the water-CO₂⁻ system. However, in the water-CO₂ system (Figure R3b), there is no overlap between CO₂ PDOS and DOS at the Fermi level. This difference can be observed at various sampling points during OS-ET simulations, as shown in Figure R4. We thank the reviewer for commenting on this.

Figure R3. Density of states (DOS) and projected density of states (PDOS) analysis in water- CO_2^- system (a) and water- CO_2 system (b). Near the Fermi level ($E-E_f = 0$ eV), the overlap between CO_2^- PDOS and total DOS is marked by yellow in a, and that between CO_2 PDOS and total DOS is marked by gray in b.

Figure R4. Density of states (DOS) and projected density of states (PDOS) analysis in water- CO_2^- system (a, b, and c) and water- CO_2 system (d, e, and f) with various sampling points during OS-ET simulations.

Modifications: The additional discussion and figures have been added to the supporting information to support our hypothesis as follows (supporting information Page 2, line 87~102, Figure S13 and S14):

“It is also possible that the Fermi level of the water-CO₂⁻ system is determined by CO₂⁻ rather than the surface, which defines the Fermi level in the water-CO₂ system. The analysis of density of states (DOS) and projected density of states (PDOS) supports this assertion as there is a clear overlap between CO₂⁻ PDOS and total DOS in water-CO₂⁻ system, which is not observed for the water-CO₂ system (Figure S13 and S14). These observations support the recent arguments [3] that the Fermi level is not the correct quantity for determining the electrode potential in outer-sphere reactions and that the electrode inner potential should instead be used to characterize the electrode potential.”

REVIEWER COMMENTS

Reviewer #3 (Remarks to the Author):

The authors responded satisfactorily to my main criticism and changed the paper adequately. The paper might now be published as is.

Reviewer #4 (Remarks to the Author):

Most of my comments have been addressed. There remain some issues I'll point out below.

Comment 1: The previous version of Fig 2a,d was confusing, but now the revised version has the reorganization energy labeled incorrectly. What is labeled as "lambda" on the plot is the energy gap, as it indicates an energy difference between two different diabatic states at the same point along the collective solvent coordinate. The reorganization energy is defined as the free energy difference between equilibrated and non-equilibrated solvent configurations for a given diabatic state. I don't doubt that the actual calculations are carried out correctly, but this mistake on the plot labels could confuse readers and needs to be corrected.

Comment 5: I am satisfied with the changes to the manuscript in response to my original comment, but the authors' response brings forward a few other related items that are worth clarifying.

Some of these details might need further clarification in the manuscript. For example, ΔA is a reaction free energy (i.e., a free energy difference between reactant and product ground states) so I don't see how it would depend on an energy gap, as indicated in the Methods and the authors' response. So, it remains unclear why this would depend on whether the linear response, or non-linear forms, of Marcus theory are used.

Some of the comments the authors have made about uncertainty in the definition of the two diabatic states are important, and would be useful to the community to include in the manuscript.

REVIEWER COMMENTS

Reviewer #3 (Remarks to the Author):

The authors responded satisfactorily to my main criticism and changed the paper adequately. The paper might now be published as is.

Response: We really appreciate the reviewer for previous rounds' comments and feedback, and we acknowledge all the efforts from the reviewer helping to improve our manuscript.

Reviewer #4 (Remarks to the Author):

Most of my comments have been addressed. There remain some issues I'll point out below.

Response: We sincerely thank the reviewer for raising these issues in our last round response, and we fully agree further clarifications are needed to improve our manuscript. Based on the following comments, specifically in comment 1, we modified the Figure 2 for the schematic illustration of Marcus theory, where we made a stupid mistake by labeling "lambda" incorrectly. In comment 5, we try our best to discuss the linear/non-linear response of Marcus theory, and we also made further discussion on why the reaction energy is dependent on the energy gap. We agree with the reviewer that it is important to discuss the uncertainty in the definition of the two diabatic states, which has been added into the manuscript. For the linear/non-linear response of Marcus theory, we think it might be a bit technical for our current work, and we added a brief clarification in the manuscript (this full discussion was added into the supplementary information). We would appreciate if the reviewer has any suggestions on this. We would like to thank the reviewer again for the encouraging comments and valuable suggestions to improve the quality of our manuscript.

Comment 1: *The previous version of Fig 2a,d was confusing, but now the revised version has the reorganization energy labeled incorrectly. What is labeled as "lambda" on the plot is the energy gap, as it indicates an energy difference between two different diabatic states at the same point along the collective solvent coordinate. The reorganization energy is defined as the free energy difference between equilibrated and non-equilibrated solvent configurations for a given diabatic state. I don't doubt that the actual calculations are carried out correctly, but this mistake on the plot labels could confuse readers and needs to be corrected.*

Response: We thank the reviewer for pointing this sloppy drawing in the previous figure. The reviewer is correct that the reorganization energy is the free energy difference between equilibrated and non-equilibrated solvent configurations for a given diabatic state. We corrected the figure as shown in Figure R1, and we also clearly marked the adiabatic ground state, adiabatic excited state, and diabatic states to help

readers to identify the schematic figure of Marcus theory used in our work.

Modifications: The modified Figure 2a and 2d are shown below and corresponding changes are made in Figure 2 within the manuscript (Page 6).

Figure R1. Schematic view of the Marcus theory in Au-water (a) and Au-water-2K (d). The orange (blue) solid line is the adiabatic ground (excited) state while the black dashed lines are the two diabatic states (A, B) describing the initial and final states of CO₂ and CO₂⁻, respectively.

Comment 5: *I am satisfied with the changes to the manuscript in response to my original comment, but the authors' response brings forward a few other related items that are worth clarifying.*

Some of these details might need further clarification in the manuscript. For example, ΔA is a reaction free energy (i.e., a free energy difference between reactant and product ground states) so I don't see how it would depend on an energy gap, as indicated in the Methods and the authors' response. So, it remains unclear why this would depend on whether the linear response, or non-linear forms, of Marcus theory are used.

Some of the comments the authors have made about uncertainty in the definition of the two diabatic states are important, and would be useful to the community to include in the manuscript.

Response: We thank the reviewer for these comments. These concerns can be divided into two parts: (1) why the reaction free energy depends on the energy gap and why this depends on linear or non-linear forms of Marcus theory; (2) our discussions (in last-round response) on the uncertainty in the definition of two diabatic states are important which should be included into the manuscript. We think these are really nice points and will respond as follows:

(1) Discussion and clarifications on Marcus theory, energy gap, and reaction free energy

Marcus theory can be derived in several but equally valid ways, and we agree that discussing some details, terms, and definitions used in this work is beneficial.

The linear response approximation is the central assumption behind the famous Marcus

barrier equation, $\Delta A_{Marcus}^\ddagger = \frac{(\Delta A + \lambda)^2}{4\lambda} \cdot \Delta A_{Marcus}^\ddagger$ arises only when the diabatic energy curves are harmonic along the energy gap coordinate: this condition is met when the energy gap distributions are Gaussian, which is equivalent to the linear response approximation as well as the second cumulant expansion (<https://doi.org/10.1063/1.470232>, <https://doi.org/10.1063/1.478425>). In these cases, the iconic Marcus barrier can be obtained by simulating the initial and final states only. However, in the non-linear Marcus theory, thermodynamic integration at multiple points along the reaction coordinate (energy gap) is required (<https://doi.org/10.1021/ja2069104>). Studying the non-linear effects would be very interesting, but here we restrict to the linear response theory.

In this work, we follow the cumulant expansion route based on Zwanzig's linear response theory (<https://doi.org/10.1063/1.470232>, <https://doi.org/10.1063/1.478425>), but, as stressed above, this is formally equal to the Gaussian energy gap or harmonic diabatic energy approximations. Within this linear response approximation and cumulant expansion, the reorganization energy can be computed exactly (<https://doi.org/10.1063/1.470232>, <https://doi.org/10.1063/1.478425>) from the energy gap variance σ^2 : this is equation 4a ($\lambda_I = \frac{\sigma_I^2(\Delta E)}{2k_B T}$, $\lambda_F = \frac{\sigma_F^2(\Delta E)}{2k_B T}$, $\lambda = 1/2(\lambda_I + \lambda_F)$) in Methods. Similarly, equation 4b arises from cumulant expansion and hence the linear response theory, and the reaction free energy is given in terms of the energy gap expectation value and variance through the reorganization energy. For the reviewer's reference, equation 4b to calculate reaction free energy is $\langle \Delta E_I \rangle = \Delta A + \lambda_F$, $\langle \Delta E_F \rangle = \Delta A - \lambda_I$ ($\langle \Delta E_I \rangle$ and $\langle \Delta E_F \rangle$ are the energy gap expectations). Hence, both the reaction free energy and reorganization energy depend on the energy gap distributions (expectation value and variance).

We hope our above discussions could give a better clarification on the usage of linear response Marcus theory. The reaction free energy calculation is based on the energy gap expectation value and reorganization energy (which is also derived from energy gap variance). Overall, the energy gap distributions affect the calculated reaction free energy.

Modifications: As mentioned above, the discussion about linear and non-linear forms of Marcus theory might be a bit technical for our current work, and the additional discussions are added into supplementary information. In Methods, we added a brief clarification about why the reaction free energy depends on the energy gap distributions as well as the usage of linear response form of Marcus theory in our work (Page 16, line 713~717, line 724~730).

“The reorganization and reaction free energies are then computed from the energy gap variances and expectation values as follows,[59] and both the reaction energy and reorganization energy depend on the energy gap distributions.”

“Such computations are based on the linear response form of Marcus theory. $\Delta A_{Marcus}^\ddagger$

arises only when the diabatic energy curves are harmonic along the energy gap coordinate: this condition is met when the energy gap distributions are Gaussian, which is equivalent to the linear response approximation as well as the second cumulant expansion.[60, 61] Further discussions of linear and non-linear forms of Marcus theory can be seen in the supplementary information (Supplementary S.1.3).”

(2) We agree with the reviewer that the discussion on the definition of diabatic states is important, which has been added into the manuscript.

Modifications: The following discussion is added into Page 5 line 210~216.

“It is important to note that there is no universally accepted method for defining a diabatic state. The cDFT method falls within a category of empirical valence bond methods, where the definition of diabatic states relies on chemical intuition. Here we consider the cDFT diabatic description of CO_2^- to be reasonably accurate, as evidenced by the computed vibrational frequencies of CO_2^- closely matching experimental values (Table 1).”

REVIEWERS' COMMENTS

Reviewer #4 (Remarks to the Author):

The authors have sufficiently addressed comments in previous reviews. I find the added text about nonlinear vs linear response forms of Marcus theory contain the appropriate details, while not detracting from the central points in the manuscript. One minor suggestion I have is to give consistent labels to the diabatic states (either "I" and "F" or "A" and "B"). Other than this minor comment, I find this manuscript highly suitable for publication in Nat. Comm.

REVIEWER COMMENTS

Reviewer #4 (Remarks to the Author):

The authors have sufficiently addressed comments in previous reviews. I find the added text about nonlinear vs linear response forms of Marcus theory contain the appropriate details, while not detracting from the central points in the manuscript. One minor suggestion I have is to give consistent labels to the diabatic states (either “I” and “F” or “A” and “B”). Other than this minor comment, I find this manuscript highly suitable for publication in *Nat. Comm.*

Response: We really appreciate the reviewer for previous rounds’ comments and feedback, especially for the further discussion on linear and non-linear forms of Marcus theory. We acknowledge the efforts from the reviewer helping to improve our manuscript. For the diabatic state illustration, we agree with the reviewer that it should be consistent. Therefore, we modify the figure (Figure R1) with “I” and “F” representing the two diabatic states (initial and final states).

Figure R1. Schematic view of the Marcus theory in Au-water (a) and Au-water-2K (d). The orange (blue) solid line is the adiabatic ground (excited) state while the black dashed lines are the two diabatic states (I, F) describing the initial and final states of CO₂ and CO₂⁻, respectively.

Modifications: The changes are updated in Figure 2 in the manuscript.